# Recent Advancements Towards the Use of Vitamin D Isoforms and the Development of Their Synthetic Analogues as New Therapeutics

**DOI:** 10.3390/biomedicines13041002

**Published:** 2025-04-21

**Authors:** Rajiv Patel, Harsha Kharkwal, Moumita Saha, Murugesan Sankaranarayanan, Saurabh Sharma, Subhash Chander

**Affiliations:** 1Amity Institute of Phytochemistry & Phytomedicine, Amity University Uttar Pradesh, Noida 201313, India; rpatel1@amity.edu; 2Amity Institute of Pharmacy, Amity University Uttar Pradesh, Noida 201313, India; 3Department of Pharmaceutics, School of Pharmacy, Suresh Gyan Vihar University, Jaipur 302017, India; missnandinisingh83@gmail.com; 4Department of Pharmaceutical Analysis, ISF College of Pharmacy, Moga 142001, India; mmita10718@gmail.com; 5Department of Pharmacy, Birla Institute of Technology & Science Pilani, Pilani Campus, Pilani 333031, India; murugesan@pilani.bits-pilani.ac.in; 6Department of Surgical Oncology, Stanford School of Medicine, Stanford University, Stanford, CA 94305, USA; ssharma6@stanford.edu; 7Amity Institute of Pharmacy, Amity University Haryana, Gurugram 122412, India

**Keywords:** vitamin D, vitamin D deficiency, Vitamin D Receptor (VDR), calcium homeostasis, vitamin D metabolites, vitamin D activation

## Abstract

Vitamin D and its metabolites are essential in various physiological processes, including muscle strength, metabolism, antifibrotic activity, and immune regulation. Researchers are focusing on developing vitamin D derivatives with optimized receptor selectivity and reduced systemic toxicity, enhancing their therapeutic efficacy against cancer, autoimmune disorders, and inflammatory diseases. Several analogues, such as alfacalcidol, paricalcitol, and falecalcitriol, are used for managing CKD-related bone disorders, while eldecalcitol is effective for osteoporosis, and calcipotriol against psoriasis. Recent studies have explored their impact on metabolic pathways, parathyroid hormone secretion, asthma, and liver fibrosis, revealing their broad clinical potential. Despite enormous efforts in the past decades, translations of vitamin D-drugs are disproportionately limited, mainly due to toxicity due to calcemic effects and undesirable metabolic profile. This review discusses structural modifications in vitamin D3, their influence on VDR binding, transcriptional activity, and calcium homeostasis, along with their role in targeting pathways like EGFR, KRAS, and Hedgehog in cancers. Advanced analytical techniques such as LC/ESI-MS/MS facilitate precise detection of vitamin D metabolites, further improving pharmacokinetic profiling. Future research may enable the clinical approval of novel vitamin D-based therapeutics with minimal disruption to calcium–phosphorus balance.

## 1. Introduction

Vitamin D, a secosteroid hormone, plays a crucial role in various physiological processes in the human body [1,2]. Classically, it is known to regulate blood calcium and phosphate levels, while later research has revealed that it has a role in a healthy skeleton, muscle function, immune health, nervous systems, and various cellular activities, like cell proliferation and differentiation throughout the body [3,4,5]. The lipophilic nature of Vitamin D favors its storage inside the body, preliminarily in the liver, and adipose tissue, which is released into the bloodstream to make it available for cells and tissues [6]. Recent studies have revealed that Vitamin D stored in adipose tissue has clinically significant effects on serum vitamin D even after 3 to 5 years of vitamin D supplementation [7].

Vitamin D deficiency (VDD) is a global health concern that affects a large proportion of the world’s population. According to current data, more than 1000 million people are not receiving enough vitamin D in their daily life across the world [8]. Recent data published in *Endocrine Connections* display a notable increase in the frequency of VDD over the last few decades [9]. Individuals with low sunlight exposure, such as the elderly, the homebound, those with darker skin color, vegetarians, and pregnant women including their growing fetuses, are at a higher risk of VDD [10,11,12].

Nowadays, newborns enter the world with insufficient levels of vitamin D and depend on breast milk, exposure to sunlight, or dietary supplements to fulfil their daily vitamin D requirements during their initial months of life. Since the quantity of vitamin D in breast milk depend on the mother’s own vitamin D levels, it tends to be insufficient in significant proportions of the population [13,14]. Furthermore, populations living in higher-latitude locations or following cultural norms have limited sun exposure, making them more vulnerable to vitamin D deficiency. Insufficient vitamin D levels in babies can cause skeletal abnormalities such rickets, convulsions, and respiratory problems, apart from calcium deficiency [15]. So, vitamin D supplementation has been proven effective in preventing rickets, and VDD-associated disorders in babies and children who are at higher risk owing to limited sun exposure or those with darker skin pigmentation [16,17].

Vitamin D is primarily synthesized in the skin upon exposure to sunlight, mainly by UVB radiation of wavelengths 290–315 nm [18]. It can also be obtained through dietary sources but generally too little is ingested to meet the daily requirements of an individual [19]. Once synthesized or ingested, vitamin D undergoes hydroxylation in the liver to form 25-hydroxyvitamin D (25(OH)D), which serves as the primary circulating form of vitamin D [20]. Further hydroxylation in the kidneys results in the formation of the biologically active form, 1,25-dihydroxyvitamin D (1,25(OH)2D), calcitriol [21].

### 1.1. Chemistry of Vitamin D

Vitamin D is a fat-soluble secosteroid hormone, which requires sequential hydroxylation steps for its activation. Supplementation dosage forms of Vitamin D are mainly available in the form of ergocalciferol (D2) and cholecalciferol (D3) as single-ingredient products or in combination with calcium and other micronutrients [22,23,24]. Both forms are slightly different chemically; D2 contains a double bond between carbons no. 22 and 23, and a methyl group on carbon no. 24 while D3 lacks those features (Figure 1) [25,26].

Initially produced prohormone forms of Vitamin D2 and D3 have no biological effect; the body converts them into active compounds through a two-step process. Vitamin D (both forms D2 and D3) is mostly converted in the liver into 25-hydroxyvitamin D, and is subsequently further hydroxylated in the kidneys into a metabolically active form known as 1 alpha, 25 hydroxyvitamin D, calcitriol [20,21]. Vitamin D2 is an odorless white crystalline plant and a fungal-derived secosteroid, also known as ergocalciferol or calciferol, with molecular formula C_28_H_44_O and molecular weight 396.6 g/mol [27,28]. Ergosterol, also known as provitamin D2, is, under UV radiation of wavelength 290–315, converted into previtamin D2, which under thermal isomerization is converted into vitamin D2 [29] (Figure 1).

The active metabolite of ergosterol is 1 alpha; 25 hydroxyvitamin D2 consists of 28 carbons with four distinct moieties within its chemical structure; rings A, C and D and the side chain and trienes (numbers are used to label π bonds and corresponding orbitals π1, π2, π3 and π4) are depicted in Figure 1. Ring A has been documented as being crucial for pharmacological activity [25,30].

Vitamin D3, also known as cholecalciferol, is also a white crystalline and odorless compound with molecular formula C_27_H_44_O and molecular weight 384.6 g/mol, mainly present in fatty tissue of animals [31,32].

Vitamin D3 synthesis in the skin begins with 7-dehydrocholesterol (7-DHC), a cholesterol derivative found abundantly in the plasma membranes of keratinocytes in the epidermis, particularly in the stratum basale and stratum spinosum layers [33]. Upon exposure to UVB radiation (290–315 nm), 7-DHC absorbs photons, initiating a photochemical reaction that converts it into previtamin D3, which subsequently undergoes thermal isomerization to form vitamin D3 or is further photolyzed to lymisterol and tachysterol [34].

Factors such as melanin content, aging, and the angle of sunlight influence the efficiency of vitamin D3 synthesis. Higher melanin levels and aging reduce 7-DHC availability, while the angle of sunlight affects UVB exposure. Optimal synthesis occurs when the sun is directly overhead, minimizing atmospheric UVB absorption and scattering [35,36].

Vitamin D3, once it has entered systemic circulation, is bound to a vitamin D-binding protein (DBP), which is essential for its transportation to the liver and other organs [37]. Cholecalciferol (D3) has been proven to be the preferred form for prophylaxis and treatment of vitamin D deficiencies since it enhances the level of active vitamin (calcitriol) for a long time. Calcitriol itself is short acting, and its direct use at higher doses can cause intense hypercalcemic effects, and it is also a potent inhibitor of parathyroid activity, so it is relatively less preferred for prophylaxis and treatment of vitamin D compared to cholecalciferol [38,39]

### 1.2. Receptor of Vitamin D

The Vitamin D Receptor (VDR) belongs to the steroid/thyroid hormone receptor superfamily; it has a mediating effect after binding with ligand calcitriol. VDR possesses four main domains and acts as a ligand-dependent transcription factor regulating the expression of multiple genes involved in calcium and phosphate homeostasis, cellular proliferation, differentiation, and immune modulation. Although VDR is ubiquitously expressed in diverse tissues, high expression levels are observed in classical target tissues such as the intestine, kidneys, and bone [40]. The structure of the vitamin D receptor comprises four main domains [40,41]:

Ligand-Binding Domain (LBD): Located at the C-terminus, this binds specifically to calcitriol, induces a conformational change and activates the receptor.

DNA-Binding Domain (DBD): This domain is rich in zinc finger motifs, and binds to specific Vitamin D Response Elements (VDREs) at the promoter regions of target genes.

Activation Function Domains (AF-1 and AF-2): These two domains interact with other co-regulatory proteins (co-activators or co-repressors) for modulation of gene transcription.

Heterodimerization Domain: VDR forms a heterodimer with the Retinoid X Receptor (RXR), which enhances its DNA-binding ability and transcriptional activity.

The human VDR gene contains two potential beginning locations, with a frequent polymorphism (Fok 1) converting the initial ATG start codon into ACG. This results in an alternation in the VDR protein commonly associated with lower bone density [40,42]. To explore the allosteric site of the vitamin D protein, the crystallography of the human VDR ligand-binding domain in complex phenomena with calcitriol (7QPP pdb id) was studied with the help of a molecular virtual docker (MVD 2019.7.0) using the human VDR with a molecular weight of 30.32 kDa and a resolution of 1.52 Å (www.rcsb.org, accessed on 24 June 2024). Studies identified five cavities (Figure 2A) as described in Table 1. The allosteric site of the protein was analysed at 5 Å proximity, which illustrated the first active cavity (Figure 2B). Calcitriol was found to completely fit in the active site of the vitamin D receptor with hydrogen-bonding interactions of Arg274, His305, His397, Ser237, Ser278 and Tyr143 amino acid residues [40] (Figure 2C,D).

### 1.3. Mechanism of Action of Vitamin D

Calcitriol exerts its effects through interaction with the Vitamin D Receptor (VDR) located in the cytoplasm of target cells. Upon binding calcitriol, VDR undergoes a conformational change, enabling its heterodimerization with the Retinoid X Receptor (RXR). This VDR-RXR complex then translocates into the cell nucleus, where it recognizes and binds to specific DNA sequences termed Vitamin D Response Elements (VDREs) situated in the promoter regions of target genes. Binding to VDREs serves as a molecular switch, triggering either activation or repression of gene expression depending on the specific target and cellular context. Through this cycle, VDR regulates a diverse array of physiological functions, including calcium and phosphate homeostasis, bone health, cell proliferation and differentiation, immune function, muscle function, and even mood regulation (Figure 3) [43].

### 1.4. Overview of Vitamin D and Its Modified Forms in Clinical Use

Orally administered cholecalciferol is effective in addressing vitamin D deficiency. Research studies indicate significant direct correlation between the co-administration of cholecalciferol and body calcium level with a reduced incidence of femoral neck fractures in the elderly population [44]. The Endocrine Society suggests a target range of 40–60 ng/mL for optimal health [45], while the National Institute of Health defines deficiency as levels below 20 ng/mL [46]. Other organizations may further categorize levels between 12 and 19 ng/mL as insufficiency and below 12 ng/mL as deficiency [47]. To achieve sufficient vitamin D levels, the Endocrine Society recommends daily intake ranging from 400 to 1000 IU for infants, 600 to 1000 IU for children and adolescents, and 1500 to 2000 IU for adults [48,49].

Apart from the classical use of vitamin D, required for maintaining calcium, and phosphorus homoeostasis, natural and synthetic analogues of vitamin D are clinically used as therapeutic intervention for osteomalacia, hypoparathyroidism, patients with chronic kidney disease (CKD), reversal of myopathy, psoriasis, anti-aging, and photoprotection activity [50,51,52]. Table 2 summarizes the diverse forms and synthetic analogues of vitamin D, including their use as therapeutic, prophylactic intervention and also in diverse research studies.

Among the synthetic analogues, Alfacalcidol is an analogue of vitamin D3 that acts as a prodrug for calcitriol (1α,25-dihydroxyvitamin D3). This form is one of the most potent and rapidly acting compounds currently used in the prevention and treatment of vitamin D deficiency for mineralization in bone and reversal of myopathy [50]. Doxercalciferol, or 1α-(OH)D2, is converted into the active form, 1α,25(OH)2D2, through the action of 25-hydroxylase in the liver. Clinical studies show the effectiveness of doxercalciferol in suppressing PTH levels with modest increase in serum calcium and phosphorous levels [51].

Paricalcitol, an analogue of 1,25-dihydroxyergocalciferol, is used for the prevention and treatment of secondary hyperparathyroidism associated with chronic kidney failure. This is the active form of vitamin D effective at reducing PTH concentrations without causing significant hypercalcemia or hyperphosphatemia in the body [53]. Calcipotriol is a topical synthetic vitamin D2 derivative used in the treatment of plaque psoriasis. The drug is effective in treating mild to moderate forms of psoriasis by regulating cell proliferation and differentiation. This form is reported for influencing immunologic markers with <1% activity in regulating calcium metabolism. Calcipotriol is particularly effective when used in combination with other approved drugs for treatment of psoriasis [54]. Like calcipotriol, Maxacalcitol is also used against psoriasis, having distinctly different structure, effective in modulating keratinocyte differentiation and proliferation, especially in plaque psoriasis [52].

Falecalcitriol is a newer synthetic derivative of calcitriol; it first received approval in Japan. Animal studies have shown that it is a potent inhibitor of PTH, and also it exhibited more efficacy in reducing PTH compared to alfacalcidol in clinical trials. Falecalcitriol is structurally different from calcitriol, possessing six fluorine atoms replacing six hydrogens at carbons 26 and 27. Falecalcitriol metabolism involves hydroxylation at carbon 23, while routine hydroxylation occurs at carbon 24. Falecalcitriol has one-third binding affinity to the intestinal vitamin D receptor with higher transcriptional activity compared to calcitriol [55].

Lumisterol, a human secosteroid metabolite, is synthesized from 7-dehydrocholesterol via photoisomerization in the epidermis upon UV irradiation [56]. Lumisterol plays an important role in regulating epidermal proliferation and differentiation under physiological conditions. Studies conducted on mice revealed that high doses of lumisterol significantly reduced plasma levels of 25-hydroxyvitamin D3 and calcitriol by 50 and 80%, respectively, while it increased vitamin D2 formation [57]. Lumisterol is documented for anti-aging and photoprotection activity [58].

22-Dihydroergocalciferol, also known as vitamin D4, is synthesized from precursor ergosta-5,7-dienol (22,23-dihydroergosterol) by UV radiation. Vitamin D4 has rapid action for a shorter duration, and poses little risk of hypercalcemia. Vitamin D4 is reported in limited studies to be found in some mushrooms and endophytic fungi, and in turn it is often present in plants [59]. The content of the precursor 22,23-dihydroergosterol varied across different species of mushrooms; some species like Enoki have been documented for notably higher content with an average of 16.5 mg/100 g, while other species like shiitake and chanterelle exhibit moderate levels. Oyster mushroom is documented for high content of Vitamin D4 compared to D2 [60]. Vitamin D4 is relatively less explored for therapeutic applications; it has mainly been explored in different research studies including concerning binding affinities, prevention of cancer, and cardiovascular diseases [61].

Sitocalciferol, also known as vitamin D5, is a lesser-known member of the vitamin D family, derived from precursor β-sitosterol via UV irradiation. Vitamin D5 shares structural similarities with ergocalciferol and cholecalciferol, possessing different side originates from β-sitosterol. Sitocalciferol is around 100 times less active in induction of bone-calcium mobilization and around 80-fold less active in stimulation of calcium transport from the intestine [62]. Recent studies documented vitamin D_5_ in the model plant Arabidopsis thaliana [63]. 

Researchers are exploring diverse analogues of vitamin D for other interventions in areas like anti-cancer, immune modulation, anti-inflammatory, etc. [64,65,66]. Although natural forms of vitamin D possess similar pharmacological activity to different extents but achieve therapeutic benefits, they require high doses, which generally leads to undesirable side effects, mainly hypercalcemia, including calcification of blood arteries and nephrocalcinosis [67]. A review study by Maestro emphasizes the modifications of 1,25(OH)_2_D_3_, and active forms of Vitamin D3 at its side chain, A-ring, triene system, or C-ring, alone and in combination. They also discussed the diverse nonsteroidal mimics of 1,25(OH)_2_D_3_ which possessed agonists and some antagonist activity at the VDR receptors. In this study, more than hundreds of crystal structures in the ligand-binding domain of VDR’s were discussed along with vitamin D agonist for a detailed molecular understanding of their action. This review discusses the selected vitamin D analogues presented during the past 10 years and molecular insights derived from new structural information on the VDR protein [68].

So, researchers are increasingly focused on developing such derivatives with reduced agonistic activity at the Vitamin D Receptor (VDR), ensuring better-targeted action and reduced systemic toxicity. These efforts are directed toward optimizing receptor selectivity, pharmacokinetic and pharmacodynamic properties of these derivatives, aiming for enhanced efficacy in treating a specific disease including cancer, autoimmune disorders, and inflammatory conditions. In this review, we discussed the structural modifications in the Vitamin D3 (D3) skeleton and their effects on Vitamin D Receptors (VDRs), transcriptional activity, and the inhibition of the sterol regulatory element-binding protein (SREBP), which ultimately influence serum calcium levels and bone health, including osteoporosis. The review also focuses on the anti-cancer potential of these derivatives against various cancer types, including ovarian cancer, colorectal cancer (CRC), breast cancer, and lung cancer, including the associated targets/pathways such as EGFR, KRAS, p53, tyrosine kinase, Hedgehog (Hh) pathway, etc. The outcome of studies conducted using vitamin D derivatives in combination with established drugs like 5-fluorouracil (5-FU) and tributyrin are also discussed. The study also discussed the hepatoprotective effects of derivatives against oxidative damage and other diseases like psoriasis, asthma, inflammatory bowel diseases (IBDs), and other inflammatory conditions. The review also discusses the affinity of derivatives on other allied receptors like the aryl hydrocarbon receptor (AhR), a class of ligand-activated transcription factors that regulate gene expression upon activation. Several studies focusing on the pharmacokinetic profile of the derivative and its prodrug are also covered. Further research may afford clinical approval of more derivatives against cancer and other ailments with minimum effects on the calcium–phosphorus balance.

## 2. Pharmacological Activities of Vitamin D and Its Derivatives

### 2.1. Vitamin D Derivatives as Modulator of VDR

Maekawa et al. synthesized novel derivatives of vitamin D analogue in their further study and tested antagonistic and agonistic activity along the vitamin D receptor. Several compounds were constructed by substitution of the alkyl group at 25 positions of their previous reported derivative of vitamin D and tested against VDR ligands for cell selectivity. Recombinant hVDR-LBD technology was used to evaluate the binding affinity of synthesized compounds against the VDR. Compound **1** (Figure 4) displayed the most potent activity with IC_50_ of 0.6 nM, which was 67 percent of calcitriol (IC_50_ 0.4 nM). The VDR transcriptional activity of synthesized compounds was studied, in which compound **1** showed activity with an IC_50_ value of 0.2 nM, the same as calcitriol (**2a**) (Figure 4). Testing of binding affinity with different proteins like with RXRα and SRC-1 and nuclear receptor corepressor 1 demonstrated that compound **1** displayed the highest activity with IC_50_ of 0.1 nM, 0.6 nM, and 0.4 nM, respectively, the same as natural hormone [69].

Belorusova and colleagues investigated the transcriptional activity of synthesized compounds, including **2a**, **2b**, and **2c** (Figure 4). Their findings revealed that compounds **2b** and **2c** exhibited approximately fourfold higher luciferase activity in response to several human point-mutated Vitamin D Receptor (VDR) variants, including hVDR Gly319Val, His305Gln, Ile268Thr, and Ile314Ser. Furthermore, in vivo studies were conducted in mice to assess the transcript levels of VDR target genes and serum calcium concentrations. The results indicated that compound **2b** induced a 1.5-fold increase in transcript levels of the duodenal VDR target genes Slc37a2 and Slc30a10 in the VDRgem model compared to the wild type. Additionally, **2b** enhanced serum calcium levels threefold in mice bearing the VDR L304H mutation compared to the vehicle-treated group [70].

Abe et al. synthesized and evaluated novel 1α,25-dihydroxyvitamin D3 derivatives featuring nitrogen-linked substituents at the A-ring C-2 position, for Vitamin D Receptor (VDR) binding affinity. In this study, small substituents with hydrogen-bonding capabilities, such as NHAc and NHMs, effectively elicited VDR transcriptional activity; in particular, derivative 2β-NHMs-1,25-VD3 (**2d**) (Figure 4) demonstrated superior activity as compared to the standard drugs 1,25-VD3. Additionally, the findings contributed in understanding how modifications at the C-2 position can alter the activity of vitamin D derivatives, potentially leading to new therapeutic applications, while substitution with the bulky group resulted in inactivation of biological activity [71].

Asano and colleagues (2016) investigated the effects of vitamin D lactam derivatives on the Vitamin D Receptor (VDR). Their previously synthesized non-calcemic vitamin D lactam derivatives (compound **3**, Figure 4) were evaluated for their ability to mimic the activity of **2a** metabolites in modulating calcium levels. The structural confirmation of compound **3** binding to the VDR ligand-binding domain was analyzed using X-ray crystallography. The results revealed significant conformational changes within the ligand-binding domain, particularly in the loop between helices H6 and H7. In particular, compound **3** demonstrated the ability to mimic the activity of vitamin D3 metabolites, inducing hypercalcemia-like effects without increasing serum calcium levels [72].

Jamali et al. assessed the role of vitamin D receptor expression by using calcitriol on the retina’s neovascularization and postnatal development of the retinal vasculature during ischemic retinopathy (OIR). The in vivo study was conducted on vitamin D receptor wild mice (Vdr +/+) and to address whether the vitamin D receptor lacks the gene (Vdr −/−); the study revealed the endothelial cells (EC)/pericytes (PC) ratio significantly decreased in P42 Vdr −/− mice as compared to Vdr +/+ mice due to an increase in pericyte (PC) density and a decrease in endothelial cell (EC) density. While there was no discernible difference between Vdr −/− and Vdr +/+ mice in terms of vessel obliteration and retinal neovascularization during OIR, VDR expression was necessary for the suppression of retinal neovascularization by compound **2a** (Figure 4) [73].

Song et al. conducted a computational study on the interaction of vitamin D derivatives with the aryl hydrocarbon receptor (AhR). Using molecular simulations, they analyzed the binding of vitamin D3 hydroxyderivatives (**4a**, **2a**, **4b**, and **4c**, Figure 4) to AhR. The derivatives stimulated the expression of AhR downstream targets cyp1A1 and cyp1B1 and induced AhR nuclear translocation, confirmed via flow cytometry and western blotting. Molecular dynamics simulations revealed that these derivatives significantly influenced the structure and dynamics of the AhR ligand-binding domain (LBD). Binding free energy values for **4a**, **2a**, **4b**, and **4c** were −23.93, −11.55, −27.52, and −17.53 kcal/mol, respectively, outperforming Indirubin (**5**, Figure 4) (−8.43 kcal/mol). The derivatives formed distinct hydrogen bonds with AhR and caused variations in residue dynamics and ligand-binding pocket properties, explaining the differing affinities of AhR for vitamin D3 derivatives and natural ligands [74].

In 2016, Biswas et al. reported the design, synthesis, and evaluation of the binding affinity of novel vitamin D derivatives toward the Vitamin D Receptor (VDR), compared to their previously reported compounds, **6a** and **6b** (Figure 4). They synthesized 24,24-difluoro-1β,3β,25-dihydroxy-19-norvitamin D3 (**6c**) (Figure 4) and 24,24-difluoro-1α,3α,25-dihydroxy-19-norvitamin D3 (**6d**) (Figure 4) and assessed their VDR binding affinity. The findings revealed that compounds **6c** (IC_50_ = 64.8 nM) and **6d** (IC_50_ = 57.6 nM) exhibited 400- and 300-fold higher binding affinities, respectively, compared to their corresponding non-fluorinated derivatives, **6a** (IC_50_ = 24.5 µM) and 6b (IC_50_ = 2.1 µM) [75].

Anami et al. investigated the ligand-binding domain (LBD) of the Vitamin D Receptor (VDR) complexed with alkyl-substituted derivatives at the 22-position of vitamin D. Their findings revealed that a single crystal structure of the complex exhibited both antagonist and agonist binding conformations, indicating the presence of mixed conformers. The partial agonist activity of the 22R-alkyl analogue was attributed to the combined effects of these antagonist and agonist conformations. This study was unique as it provided the first structural evidence of a conformational subset of the ligand and nuclear receptor within a single crystal, offering critical insights into the structural basis of partial agonism in VDR and potentially other nuclear receptors. Furthermore, the synthesized 22R-alkyl analogues were evaluated for biological activity. Compounds **7a** and **7b** (Figure 4) exhibited partial and full agonist activity, respectively, demonstrating significant binding affinity for VDR, with varying degrees of agonistic and antagonistic effects [76].

Saitoh et al. synthesized and evaluated C-2 substituted vitamin D derivatives featuring ringed side chains to investigate their effects on bone growth. These derivatives were designed with side-ring substitutions using D-glucose as a chiral template. The synthesized compounds were evaluated in vitro for their binding affinity to the Vitamin D Receptor (VDR) and their transactivation potential in human osteosarcoma (HOS) cells. The results revealed that compound 8, bearing a hydroxypropoxy side chain at the C-2 position, demonstrated strong VDR binding affinity (58%) and the highest transactivation activity, with an EC_50_ value of 6.8 pM. This compound promoted bone growth while potentially exhibiting lower calcemic activity compared to other derivatives. These findings suggest that compound **8** holds promise for therapeutic applications aimed at improving bone health. Additionally, the study highlighted the structure–activity relationship (SAR) of C-2 position modifications in vitamin D3 derivatives, indicating the potential of compound **8** (Figure 4) for future use in bone-growth-enhancement therapies [77].

### 2.2. Vitamin D Derivatives as Anticancer Agents

Maj and colleagues reported the anticancer elements such as tyrosine kinase inhibitors (imatinib (**9**, Figure 5) and sunitinib (**10**, Figure 5)) in combination with vitamin D derivatives against lung cancer cell lines (A549 and HLMECs) and antiangiogenic activities of cytostatic drugs, cisplatin (**11**, Figure 5). In vitro studies demonstrated that tyrosine kinase inhibitors exhibited distinct mechanisms of action: imatinib induced caspase-dependent cytotoxicity, while sunitinib acted through a caspase-independent pathway. Compounds **12a** and **2a** (Figure 5) exhibited synergistic effects specifically in HLMECs (human lung microvascular endothelial cells). Furthermore, tyrosine kinase inhibitors, such as sunitinib and cisplatin, were found to inhibit VEGF-A secretion from the A549 lung cancer cell line via p53-independent and p53-dependent mechanisms, respectively. The in vivo anticancer activity of selected drugs in combination therapy was also evaluated. Notably, compound **12a** demonstrated a remarkable synergistic anticancer effect when combined with tyrosine kinase inhibitors, achieving a mean tumor mass of 0.271 g. This effect was most pronounced in a triple combination therapy involving sunitinib and docetaxel (**13**, Figure 5) in the A549 cell line model [64].

Wierzbicka et al. investigated the association between colorectal cancer (CRC) and vitamin D3, emphasizing the growing link between CRC and unhealthy lifestyle factors. Low serum levels of vitamin D3 have been correlated with an increased risk of CRC, while supplementation has been suggested to reduce this risk. Vitamin D3 is known to inhibit cancer cell proliferation; however, excessive intake may lead to hypercalcemia. The study aimed to compare the antiproliferative effects of classical vitamin D3 metabolites (**2a** and **12b**, Figure 5) with low-calcemic analogues (**12c** and **4c**, Figure 5) on CRC cell lines, while also examining the expression of vitamin D-related genes in CRC tissue samples. Key findings revealed that vitamin D3 analogues, particularly **12c** (Figure 5) demonstrated significant antiproliferative activity against CRC cells (HCT116), with an IC_50_ value of 5.3 nM (MTT assay), compared to **2a** (Figure 5) (IC_50_ = 47 nM), as well as inhibition of colony formation. Gene-expression analysis highlighted differential sensitivities of CRC cell lines to these analogues. The clinical implications suggest that low-calcemic vitamin D3 analogues, such as **12c**, can show promise as potential therapeutic agents for CRC. Moreover, gene-expression profiling may aid in the personalization of anticancer therapies, thereby improving therapeutic outcomes for CRC management [78].

Milczarek et al. investigated the antitumor effects of vitamin D analogues **12d** and **12e** (Figure 5) in combination with 5-fluorouracil (5-FU, compound **14**) for the treatment of human colon cancer (HT-29). In vitro studies demonstrated that the combination of vitamin D analogues with 5-FU exhibited superior anticancer activity compared to using only one. Proliferation inhibition ranged from 1.6% to 49.9% when the analogues were used alone and from 50.9% to 54.1% when used in combination. Notably, **12e** enhanced the proportion of cells in the sub-G1 phase while reducing the proportion of intestinal tumor cells in the G2/M and S phases. Further analysis revealed that combination therapy with **12d** and compound **13** (Figure 5) in mice led to increased expression of the Vitamin D Receptor (VDR). Compared to docetaxel alone, the combination therapy resulted in decreased phosphorylation of ERK1/2 (p-ERK1/2) and increased expression of p21 in in vitro experiments. Interestingly, the pro-apoptotic effects of docetaxel were mitigated by the concurrent use of **12d**, yet **12d** still exhibited a significant therapeutic effect by slowing tumor growth despite these unfavorable apoptotic interactions. The findings suggest that the anticancer potentiation mechanisms of **12d** and **12e** involve increased p21 expression and decreased p-ERK1/2 levels, potentially leading to reduced thymidylate synthase expression. The relatively low toxicity and sustained anticancer efficacy of **12e** may be attributed to its higher binding affinity for both the vitamin D-binding protein (DBP) and VDR, as well as for the CAR–RXR ligand-binding domain [79].

In another study, Milczarek et al. examined the pharmacological impact of combination therapy utilizing compound **13** (Figure 5) and a vitamin analogue. The synergetic effect of **13** with different vitamin D analogues, such as **12d** (Figure 5) or 5,6-trans-isomer of **12c** (**12e**, Figure 5), was evaluated for treatment of colon cancer for the first time in an in vivo model system. The results showed that the combination therapy of compound **13** with **12d** inhibited 96 percent TG (tumor growth) and 21 percent decreased in body weight. It was noted that the anticancer activity of **13** was greatly increased by both **12d** and **12e**; however, the outcomes were contingent upon the treatment schedule. The combination therapy showed considerable reduction in tumor development and metastasis as well as an extension of the mice’s life period as compared to when **13** was administered alone. Neither combination was hazardous and both showed signs of a synergistic impact. Moreover, the anticancer impact of the medication was extended when analogues were applied after the **13** courses of therapy had finished. Moreover, potentiation of compound **13**’s action was noted with administration of prodrug capecitabine [80].

Maj et al. assessed the activity of vitamin D analogues against EGFR, KRAS, p53 mutation status, and VDR polymorphism in the treatment of lung cancer. With different EGFR, KRAS, p53 mutation status, and VDR polymorphism were susceptible to some vitamin D derivatives (VDDs) with antiproliferative effects. The chosen cell lines showed varying responses to VDDs. While KRAS and/or p53 mutant cells showed a lesser response, EGFR mutant cells showed significantly less cell growth. The most active compound 24,24-Dihomo-1,25D3 (**12f**, Figure 5) displayed the best anticancer activity in VDD-sensitive cell lines, such as A549, HCC827, NCI-H1299, and NCI-H1703, as compared to other tested molecules with 21.72, 40.2, 28.75, and 39.30% inhibition, respectively. Compound **12f** was selected as the primary vitamin D derivative (VDD) for further structural optimization. However, none of the VDDs demonstrated antiproliferative activity against the A-427 and Calu-3 cell lines. It was observed that the less transcriptionally active form of the Vitamin D Receptor (VDR) was present in HCC827 cells, which were sensitive to VDDs, while the more transcriptionally active form of VDR was found in NCI-H358 cells, which were stimulated to proliferate by VDDs. This led to the conclusion that VDR polymorphisms are inversely correlated with sensitivity to the antiproliferative effects of VDDs. The heightened antiproliferative activity of VDDs in HCC827 cells may be attributed to the absence of KRAS and/or p53 mutations, while the presence of these mutations in other cell lines may negate the antiproliferative effects of VDDs, despite their transcriptional activity as evidenced by increased CYP24A1 expression. Importantly, the susceptibility of the examined cells to VDDs does not appear to be directly caused by polymorphisms in the VDR gene [81].

Wang et al. (2005) identified the active metabolites of vitamin D analogues and tested their anticancer activity without increasing the level of calcium. The metabolism of **12g** (Figure 5) in HepG2 cells was evaluated, which suggested the active metabolite of **12g** was **12h** (Figure 5), as was identified by HPLC and spectral analysis. Further, anticancer activity of active metabolites of **12g** was evaluated against LNCaP (prostate cancer) and MCF-7 (breast cancer) cell lines. The finding results demonstrated that growth inhibition was observed with 60.5 ± 3.4% in LNCaP and 65.4 ± 1.3% in MCF-7 cells that were dose-dependent in **12h**; however, a comparable level of inhibition needed a concentration five times higher than that of calcitriol. Additionally, the anticancer activity of metabolite **12h** along with chemotherapy medications was also investigated. The data revealed that compound **12h** increased potency of suppression of LNCaP and MCF-7 cell growth when combined with doxorubicin and genistein. These valuable studies imply that **12h** may have therapeutic value in the management of breast and prostate cancers [82].

DeBerardinis et al. investigated the anticancer activity of vitamin D3-based derivatives incorporating aromatic A-ring mimics designed to inhibit the Hedgehog (Hh) signaling pathway. A series of vitamin D3 analogues was evaluated for their inhibitory effects on the Hh pathway using both Hh-dependent mouse embryonic fibroblasts and cultured cancer cell models. The study findings demonstrated that aromatic A-ring mimics containing one or more hydrophilic substituents on a six-membered ring, such as compound **5**, effectively blocked the Hh pathway in cultured cancer cells, with IC_50_ values ranging from 0.74 to 10 μM. Further investigations were conducted to elucidate the molecular mechanisms through which the most potent analogues, **12i** (Figure 5) and candidate **16** (Figure 5), inhibited Hh signaling. The results indicated that while the anti-Hh activity of **12i** is predominantly mediated through vitamin D receptors, candidate **16** exerts its inhibitory effects on the Hh pathway via a different mechanism [83].

Liu et al. evaluated the combined effect of vitamin D (VD), compound **17** (Figure 5)-based photodynamic therapy (ALA-PDT) and paclitaxel (**18**, Figure 5) in the cancer treatment. The results showed that breast cancer cells treated with compound **18** or ALA-PDT alone showed dose-dependent inhibition of viability, inhibition of migration and invasion, increased apoptosis, down-regulation of Bcl-2 expression, up-regulation of Bax expression, and cleavage of caspase-3. ALAPDT and compound **18** in combination showed more pronounced impacts on the parameters listed above. Furthermore, the combination also inhibited the development of xenograft tumors and the expression of Bcl-2, but increased the expression of Bax and cleaved caspase-3 in tumors. On the other hand, VD enhanced the combined effects of compound **18** and ALA-PDT, but had no effect on tumor development or the expression of Bcl-2, Bax, or cleavedcaspase-3 [84].

Zhang et al. designed and synthesized a vitamin D prodrug using a disulfide linker conjugated to a naphthalimide derivative. The synthesized prodrug (compound **19**, Figure 5) was subsequently evaluated for its anticancer activity and underlying mechanism of action. Compound **19** exhibited a red-shifted fluorescence within 30 min and demonstrated a highly selective detection mechanism for glutathione (GSH). Notably, the prodrug could be tracked via cellular imaging, and its anticancer activity was assessed using HEK 293T and HeLa cell lines. The results indicated that the prodrug displayed favorable biocompatibility and comparable potency to vitamin D2. In contrast, compound **20** (Figure 5) exhibited significantly lower potential for therapeutic application in living cells compared to other analogues [85].

### 2.3. Anti-Inflammatory Activity of Vitamin D Analogues

Chaiprasongsuk et al. (2020) explored the activity of CYP11A1-derived vitamin D3 products on UVB-induced inflammation and UVB-irradiated human epidermal keratinocytes. The products of CYP11A1-derived vitamin D3-hydroxyderivatives, such as **21a**, **4c**, **2a**, **4b** and **21b** (Figure 6) were tested for their anti-inflammatory and skin-protection properties in HEKn. After 24 h, pre- and post-UVB irradiation (50 mJ/cm^2^) indicated that all the CYP11A1-derived vitamin D3 secosteroids overpowered UVB-induced inflammatory responses in keratinocytes by suppressing the activity of nuclear-NF-κB-p65 and modulating IL-17, NF-κB p65, and IκB-α. The tested compounds also upregulated key markers of epidermal differentiation, likes involucrin (IVL) and cytokeratin 10 (CK10) in UVB radiation [86].

Chaiprasongsuk et al. claimed the effect of novel CYP11A1-derived analogues of vitamin D3, like **2a**, **4c**, **4a**, **4b**, and **21b** (Figure 6) and lumisterol on inflammation due to UVB radiation in human keratinocytes via stimulation of Nrf2 and P53 defense mechanisms. All tested compounds including vitamin D derivatives and lumisterol significantly decreased the formation of oxidants. Additionally, cell proliferation of keratinocytes and DNA damage were tested in derivatives; they blocked the proliferation at 100 nM concentration and also repaired the DNA damage by altering the 6-4PP and CPD levels with antioxidant properties [87].

Piotrowska and team explored the cytotoxic effect of hydrogen peroxide (H_2_O_2_) and cisplatin (**11**) in combination with vitamin D analogues on human keratinocyte cells. Vitamin D analogues, **2a**, **4c**, **21c**, and calcipotriol (Figure 6) were evaluated for cytotoxicity activity on HaCaT keratinocytes and oxidative stress properties with H_2_O_2_ compounds. The finding results demonstrated that vitamin D derivatives significantly suppressed the growth of the HaCaT keratinocytes with IC_50_ values ranging between 0.081 and 1.44 nM. The oxidative stress property of H_2_O_2_ was reduced up to 3.8 times (IC_50_ = 49 nM) with a combination of **21c** as compared to single H_2_O_2_ (IC_50_ = 186 nM) [88].

Martinesi et al. (2014) investigated the effects of vitamin D derivatives (**2a** and **21d**, Figure 6) on adhesion molecules and matrix metalloproteinases (MMPs) in the intestinal tissue of patients with inflammatory bowel diseases (IBDs). Biopsies from both inflamed and non-inflamed regions of the intestine, as well as peripheral blood mononuclear cells (PBMC) from IBD patients, were cultured with or without vitamin D derivatives. The researchers measured MMP activity and levels of adhesion molecules. The results indicated that vitamin D derivatives such as **2a** and **21d** significantly decreased the protein levels of ICAM-1 and MAdCAM-1 in both inflamed and non-inflamed regions. Additionally, there was a reduction in the expression of MMP-9, MMP-2, and MMP-3. The study suggested that vitamin D derivatives **2a** and **21d** may be effective in managing IBDs by targeting adhesion molecules and MMPs, highlighting their potential therapeutic role in this condition [89].

Lin et al. (2018) reported the synthesis of novel derivatives of 20S-hydroxyvitamin D3 and its 1α-OH form and subsequently tested for Vitamin D Receptor (VDR) agonistic and anti-inflammatory activity. Among the tested analogues, compound **21e** (Figure 6) significantly enhanced the expression of vitamin D target genes. VDR activity was evaluated in three cell lines, including Caco-2, HaCaT, and Jurkat. The results demonstrated that compound **21f** (Figure 6) exhibited significantly higher activity compared to other synthesized derivatives and showed notable efficacy relative to the standard compounds **2a**, **22-Oxa**, and **4c** (Figure 6). Furthermore, anti-inflammatory studies revealed that compound **21f** remarkably inhibited the production of IFNγ, with a reduction ratio of 0.381, making it the most potent anti-inflammatory agent compared to the standard drugs [66].

Abdul-Wahab et al. (2020) evaluated the in vivo effect of vitamin D3 on methotrexate (**22**, Figure 6)-induced jejunum damage in rats. Malondialdehyde content (MDA) and Total Antioxidant Capacity (TAOC) was evaluated after taking vitamin D3 for 21 days and one dose of compound **22** on day 17. There was no significant change (*p* > 0.05) in MDA, but there was a significant decrease (*p* < 0.05) in the TAOC level in the jejunum tissue. In addition, there was severe damage to the villi, an abscess in the crypt, atrophy of the epithelium, infiltration of mixed inflammatory cells, and depletion of goblet cells when compared to the methotrexate group. The study recommended the combined use of vitamin D3 and methotrexate, which not only may overcome dose restrictions and adverse effects but also possessed synergistic impact when employed for the treatment of cancer, rheumatoid arthritis, and psoriasis [90].

### 2.4. Dual Activities (Binding Affinity and Anti-Cancer) of Vitamin D Derivatives

Kudo et al. investigated the binding affinity of vitamin D derivatives toward the Vitamin D Receptor (VDR) by modifying the side chain of vitamin D. Stereoselective derivatives were synthesized through the substitution of an adamantane ring at positions 25 and 26, and the introduction of a triple bond at position 23 on the vitamin D side chain. The synthesized compounds were evaluated for their VDR binding affinity using competitive binding assays with 1,25-(OH)_2_D_3_ and recombinant Hvdr-LBD. The results demonstrated that compound **23** (Figure 7) exhibited remarkable activity with an IC_50_ of 0.5 nM, achieving approximately 90% of the binding efficiency of calcitriol. Further evaluation using a luciferase reporter assay to study VDR transactivation in HEK293 (human kidney cell lines) revealed that compound **23** was the most potent, with an IC_50_ of 0.07 nM, surpassing the other three derivatives and achieving 81% of the natural hormone’s activity. The binding affinity of VDR to RXRα, as activated by the synthesized compounds, was also assessed. Notably, compound **23** demonstrated superior activity, with an IC_50_ of 0.6 nM, outperforming the standard calcitriol (IC_50_ of 0.7 nM). Furthermore, compound **23** enhanced the binding affinity of VDR to SRC-1, showing greater potency compared to the natural hormone (IC_50_ of 1.9 nM). These findings highlight compound **23** as a promising candidate with superior VDR binding affinity and transactivation potential, outperforming both the natural hormone and other derivatives tested [91].

Slominski and colleagues synthesized and characterized CYP11A1-derived hydroxy-substituted vitamin D derivatives, which were subsequently evaluated for their anticancer potential against basal cell carcinoma (ASZ001) and squamous cell carcinoma (A431 and SCC-13) models. The synthesized derivatives demonstrated affinity for the Vitamin D Receptor (VDR) while acting as inverse agonists for retinoid-related orphan receptors (RORα and RORγ). The anticancer activity of the most potent vitamin D derivatives was assessed, in which compounds **2a**, **1**, and **24a** (Figure 4 and Figure 7) significantly suppressed GLI1 and β-catenin expression in ASZ001 cells while promoting involucrin expression in A431 cells. Additionally, the CYP11A1-derived derivatives were found to enhance their therapeutic potential by modulating genes involved in vitamin D metabolism. These findings suggest CYP11A1-derived vitamin D derivatives as promising candidates, which can be explored further [92].

Piatek et al. designed, synthesized, and evaluated vitamin D analogues against the ovarian cancer cell line as well as regulation of the vitamin D system. The four synthesized candidates were tested on two HGSOC (high-grade serous ovarian cancer) cell lines with 13,781 (low) and 14,433 (high) CYP24A1 mRNA gene-expression levels. The results demonstrated that the candidate **24b** (Figure 7) displayed high activity in both cell lines with EC50 values of 2.98 nmol/L in 13,781 cells and EC50 = 0.92 nmol/L in 14,433 after 4 h. After 5 days of continuous treatment, **24b** had more therapeutic effect in low cell line 13,781 as compared to high cell line 14,433. In cellular vitamin D receptor binding studies, test compounds showed good interaction and high nuclear vitamin D receptor levels in only 13,781 cells. Furthermore, cell viability of all synthesized vitamin D derivatives was evaluated, which revealed that all tested candidates decreased the cell viability [93].

In 2018, Wallbaum et al. worked on the effects of vitamin D analogues, including vitamin D2 (**24c**, Figure 7), vitamin D3 (**12i**, Figure 7), and calcipotriol (**12c**, Figure 7) in pancreatic stellate cells (PSCs). In vitro effects of vitamin D analogues on cultures of PSC were explored; all tested analogues significantly inhibited the growth of stress fiber bundles in primary cultures of murine PCS with a score α-SMA value of 1.23 (**12c**), 1.40 (**24c**), 1.58 (**12i**), 2.54 (control), while they were inactive in re-cultures of PCS. Additionally, researchers also studied the effects of vitamin D derivatives on DNA synthesis of PSCs, in which no changes in DNA formation were observed. Further, gene expression and secretion evaluation displayed remarkable inhibition of mRNA and protein levels of IL-6, along with increasing the expression of the VDR target gene [94].

Mizumoto et al. synthesized a series of compounds with substitution of the alkoxy group at C2 position of 19-nor vitamin D3 and further evaluated activity against the HL-60 cell line. The findings of this study demonstrated that methoxy containing compound **25** (Figure 7) showed the most potent activity against HL-60 with ED_50_ 0.38 nM, which was 26 times more compared to the reference drug calcitriol. Additionally, binding affinity of synthesized candidates was performed within the vitamin D receptor [95].

Kawagoe et al. (2021) designed and synthesized a series of vitamin D analogues featuring substitutions in the pseudo A-ring to selectively inhibit the sterol regulatory element-binding protein (SREBP) without activating the Vitamin D Receptor (VDR) or disrupting calcium homeostasis. A total of 50 compounds were synthesized and evaluated in vitro as SREBP inhibitors in cell culture, ensuring no interference with VDR activity. The results demonstrated that **compound 26** (Figure 7) exhibited the highest inhibition of SREBP, with an IC_50_ value of 0.7227 µM, comparable to the reference compound **12b** (Figure 7) (IC_50_ = 1.176 µM). Furthermore, **compound 26** was assessed for its efficacy and safety in treating fatty liver in mice. After four weeks of treatment, the mice administered compound **26** remained healthy, whereas those treated with the control drug **12b** exhibited signs of illness due to hypercalcemia caused by excessive calcium levels [96].

Brożyna et al. investigated the anticancer properties of vitamin D and its hydroxy derivatives against ovarian cancer cells. The study focused on the phenotypic effects of hydroxy derivatives of vitamin D and their agonistic activity on Retinoic Acid-Related Orphan Receptors (ROR-γ/α) in ovarian cancer cell lines. The findings revealed that the Vitamin D Receptor (VDR) and RORs were localized in both the nucleus and cytoplasm, with their expression levels significantly reduced in primary and metastatic ovarian cancers compared to normal ovarian epithelium. In SKOV-3 and OVCAR-3 cell lines, however, the expression levels of VDR and ROR-γ/α were notably upregulated. The anticancer activity of the selected compounds was evaluated in various cancer cell lines, including SKOV-3 and OVCAR-3. Compounds **27** and **28** (Figure 7) effectively inhibited the growth of OVCAR-3 cells, while compound **4c** (Figure 7) suppressed the growth of SKOV-3 cells. Additionally, compounds **2a**, **27**, and **28** significantly inhibited spheroid formation in SKOV-3 cancer cell lines, which reveals their potential therapeutic applications in ovarian cancer treatment [97].

Rhieu et al. investigated the pharmacokinetics of a synthetic derivative of compounds **2a** (Figure 7) and 29 in a cell-free system. The metabolism of compound **29** (Figure 7) was assessed, and the findings revealed that neither cytochrome CYP27B1 nor cytochrome P450 24A1 (CYP24A1) exhibited any effect on it. This indicates that the structural modifications in compound **29** enabled the analogue to resist the metabolic actions of both CYP27B1 and CYP24A1. These observations were further validated in immortalized normal human prostate epithelial cells (PZ-HPV-7), which are known to express both CYP27B1 and CYP24A1. The authors also conducted molecular docking studies between the analogue and CYP27B1 to provide detailed insights into the structure–function relationship. Additionally, it was found that compound **29** exhibited potency comparable to compound **2a** in suppressing the proliferation of PZ-HPV-7 cells. The antiproliferative activity of compound **29** was determined to be dependent on the Vitamin D Receptor (VDR), as it failed to inhibit the proliferation of breast tumor cells derived from animals lacking VDR. Furthermore, the growth-inhibitory effect of compound **29** was restored following the stable introduction of VDR into VDR-knockout cells. Overall, the study identified compound **29** as a novel non-1α-hydroxylated vitamin D analogue with antiproliferative activity equivalent to that of compound **2a** [98].

Gaschott et al. investigated the effects of tributyrin (**30**, Figure 7) on the expression of the Vitamin D Receptor (VDR), as well as the proliferation and differentiation of the human colon cancer cell line Caco-2. The anticancer activity of prodrug **30** and its synergistic effects with compound **2a** (Figure 7) were evaluated against Caco-2 cells. The results demonstrated that prodrug **30** alone exhibited greater potency in suppressing Caco-2 cell proliferation (22–53%) compared to natural butyrate (11–23%) and compound **2a** (8–15%). Notably, in combination with compound **2a**, the suppression of proliferation increased to 55%. The addition of physiological concentrations of dihydroxycholecalciferol further amplified this effect significantly. RT-PCR analysis revealed that the tributyrin-induced upregulation of the vitamin D receptor contributed to the observed synergistic effect of prodrug **30** (Figure 7) and compound **2a** (Figure 7) in Caco-2 cells. Treatment with compound **30** resulted in a 1.5-fold increase in the binding of compound **2a** to its receptor, although it did not alter the receptor’s affinity [99].

Fujii et al. conducted a study focused on the design and synthesis of tetraol derivatives of 1,12-dicarba-closo-dodecaborane as non-secosteroidal vitamin D analogues. The key findings revealed that introducing an ω-hydroxyalkoxy functionality significantly enhanced the biological activity of these compounds against cancer cells (HL-60). Among the derivatives, the 4-hydroxybutoxy non-secosteroidal vitamin D analogue emerged as the most potent, exhibiting an IC_50_ value of 24 nM for inhibiting HL-60 cell proliferation, compared to the reference compound (IC_50_ = 74 nM). The significance of these compounds lies in their potential therapeutic applications targeting the Vitamin D Receptor (VDR), which plays a critical role in bone metabolism and immune response. The study also referenced numerous prior investigations into VDR ligands and their biological activities, highlighting the relevance of this research in advancing the field of vitamin D-based therapeutics [100].

### 2.5. Vitamin D Analogues as Potential Anti-Osteoporotic Agents

Sato et al. (2014) investigated the effects of vitamin D derivatives, including compounds **2a** and **32a** (Figure 8), on bone health in the context of osteoporosis treatment. An in vitro study on osteoclast activity revealed that compound **32a** exhibited superior inhibition of osteoclast activity by suppressing HIF1α protein, a key regulator of osteoclast development, compared to the vitamin D metabolite **2a**. Furthermore, the study demonstrated that compound **32a** exhibited minimal activity against the macrophage colony-stimulating factor (M-CSF) and receptor activator of nuclear factor kappa B ligand (RANKL), both critical for osteoclastogenesis. The findings indicated that compound **32a** significantly reduced osteoclast activity compared to compound **2a**. The study emphasized targeting HIF1α using the vitamin D derivative as a therapeutic strategy for postmenopausal osteoporosis [101].

Takeda et al. (2017) investigated the efficacy of the vitamin D derivative compound **32a** (Figure 8) in mitigating trabecular bone loss associated with diabetes. The study evaluated compound **32a** as a potential agent for preventing diabetes-induced bone loss in streptozotocin-induced type I diabetic rats. Treatment regimens included compound **32b** (Figure 8) (25, 50, or 100 ng/kg), vehicle, or compound **32a** (10, 20, or 40 ng/kg), administered five times per week for 12 weeks, starting one week after diabetes induction. The results revealed that serum osteocalcin (OC) levels (~10 ng/dL) remained unaffected by any dosage of compound **32a** or the highest dose of compound **32b**. However, urinary deoxypyridinoline (DPD) excretion was significantly reduced. Compound **32b** showed no notable effects on these parameters, whereas compound **32a** at 20 and 40 ng/kg doses effectively prevented bone mineral density (BMD) declines. Additionally, the highest dose of compound **32a** inhibited reductions in the maximum load of the lumbar vertebrae. Despite these observations, the biomechanical strength and areal BMD of the femoral shaft were not significantly affected by either compound. Diabetic rats exhibited lower bone volume and trabecular thickness in the lumbar vertebrae, as well as increased trabecular separation, compared to non-diabetic controls. Treatment with both compound **32a** and compound **32b** prevented further deterioration of the trabecular microstructure. However, advanced glycation end product (AGE) content in the femoral cortical bone was higher in diabetic rats, irrespective of treatment, and neither compound affected AGE accumulation through independent processes [102].

Kato et al. (2015) investigated the effects of compound **2a** (Figure 8) in optimizing a novel method for purifying induced pluripotent stem (iPS)-derived osteoprogenitors. The study evaluated the role of growth factors, including fibroblast growth factor 2 (FGF-2), insulin-like growth factor 1 (IGF-1), and transforming growth factor-beta (TGF-β). The results demonstrated that early-phase osteoblasts emerged after 10 days of treatment, while late-phase osteoblasts were observed after 14 days. Further investigation into the effects of vitamin D analogues revealed that compound **2a** significantly enhanced the expression of osteocalcin while simultaneously reducing the expression of tissue-non-specific alkaline phosphatase (TNAP) and runt-related transcription factor 2 (RUNX2). These findings suggest that treatment with compound **2a** (Figure 8) promotes the differentiation of iPS-derived day-14 osteoprogenitor cells (iPSop-day14) into mature osteoblasts [103].

Takeda et al. (2015) evaluated the potential of vitamin D analogues, such as compound **32a** (Figure 8), to regulate bone metabolism during long-term treatment in ovariectomized (OVX) rats. Compound **32a** was administered orally to six-month-old Wistar-Imamichi rats at doses of 7.5, 15, or 30 ng/kg per day following ovariectomy. Bone mineral density (BMD), urinary excretion of deoxypyridinoline (DPD) as a marker of bone resorption, and serum total alkaline phosphatase (ALP) as an indicator of bone formation were measured at 3, 6, and 12 months of treatment. After 12 months, bone histomorphometry of the tibial diaphysis and L3 lumbar vertebra, as well as biomechanical strength assessments of the femoral shaft and L4 lumbar vertebra, were performed. The results demonstrated that compound **32a** effectively prevented OVX-induced reductions in BMD in both the femur and lumbar vertebrae throughout the treatment period. Furthermore, compound **32a** significantly reduced OVX-induced increases in urinary DPD excretion, with minimal effects observed for compound **32b** (Figure 8). Importantly, compound **32a** preserved the biomechanical properties of bone, preventing the OVX-induced decreases in ultimate load and stiffness of the L4 lumbar vertebra and femoral shaft. Histomorphometric analysis of the L3 lumbar vertebra revealed that 12 months of treatment with compound **32a** suppressed OVX-induced increases in bone-resorption parameters (osteoclast surface and osteoclast number) and bone-formation parameters (osteoblast surface, osteoid surface, and bone-formation rate). Additionally, compound **32a** restored activation frequency, which was elevated in the OVX/vehicle group, to baseline levels, indicating that it maintained bone turnover at physiological levels. Moreover, compound **32a** effectively prevented OVX-induced deterioration of the trabecular and cortical bone microstructure. Overall findings of the study highlight the potential of compound **32a** as a therapeutic agent for maintaining bone health and preventing osteoporosis-related bone loss [104].

Ringe et al. investigated the efficacy of compound **32b** (Figure 8) in the treatment of male osteoporosis through a two-year clinical trial involving 214 patients. The study design included two groups: Group B (control group) received 1000 IU of standard vitamin D (Vit. D) plus 1000 mg of calcium daily, while Group A, comprising patients at higher risk of incident fractures, was administered 1 µg of compound **32b** combined with 500 mg of calcium. An additional 56 patients (28 with and 28 without prevalent vertebral fractures) were later included in the study. After two years, Group A demonstrated significantly greater improvements in bone mineral density (BMD) at the total hip (+1.9% vs. −0.9%) and lumbar spine (+3.2% vs. +0.8%) compared to Group B. Furthermore, the incidence of falls was significantly lower in the compound **32b** group (18 falls) compared to the vitamin D group (38 falls; *p* = 0.041). Group A also exhibited a significantly reduced occurrence of new vertebral and non-vertebral fractures compared to Group B. Notably, compound **32b** was associated with a significant reduction in back pain, a benefit not observed in the control group. A correlation was identified between renal function and the incidence of new non-vertebral fractures, with compound 32b demonstrating superior efficacy in reducing fractures among patients with a creatinine clearance (CrCl) of less than 60 mL/min (*p* = 0.0019). The safety profile of compound **32b** was comparable to that of standard vitamin D, with no significant differences in the frequency of mild-to-moderate adverse events (AEs) between the groups and no reported serious adverse events (SAEs). Despite a higher baseline fracture risk in the compound **32b** group, the two-year treatment with this active D-hormone prodrug resulted in superior therapeutic outcomes, including enhanced BMD, reduced falls, and fewer fractures [105].

Kashiwagi et al. (2013) designed, synthesized, and biologically evaluated a series of nonsecosteroidal Vitamin D Receptor (VDR) agonists to assess their therapeutic potential in addressing osteoporosis and bone mineral density (BMD) loss. The study focused on gamma hydroxy carboxylic acid analogues, where the authors replaced the 1,3-diol moiety of compound **2a** (Figure 8) with aryl acetic acid derivatives to develop novel nonsecosteroidal VDR agonists. In vitro evaluation of the synthesized compounds revealed that the tested analogues exhibited significantly higher potency compared to compound **2a**. X-ray crystallographic analysis of compound **33** demonstrated that while its carboxylic acid interacted with the VDR in a distinct manner from gamma hydroxy carboxylic acids, the inserted phenyl ring closely mimicked the folded methylene linker of the gamma hydroxy carboxylic acid moiety. Subsequent in vivo studies were conducted using an osteoporosis rat model. In immature rats, compound **34** emerged as a highly potent analogue, showing a remarkable 198% increase in ΔBMD compared to compound **2a**, which exhibited only a 4% ΔBMD. In adult rats within the same model, compound **34** (Figure 8) demonstrated a favorable pharmacokinetic (PK) profile and exceptional efficacy in preventing BMD loss without inducing significant hypercalcemia. These findings suggest that the nonsecosteroidal VDR agonist **34** holds promise as a novel therapeutic candidate for the treatment of osteoporosis in humans, offering a potential alternative to traditional secosteroidal vitamin D analogues [106].

### 2.6. Metabolic Studies of Vitamin D Derivatives

In 2021, Sakamoto and colleagues investigated various analogues of side chain hydroxy-substituted cholecalciferol. The synthesized compounds were evaluated to determine the role of the CYP27B1 protein in drug metabolism. Their findings demonstrated that CYP27B1 is responsible for metabolizing the synthesized analogues into their corresponding derivatives, including **35a**, **35b**, **35c**, **35d**, **35e**, and **35f** (Figure 9) marking the first report of this metabolic pathway. CYP27B1 exhibited a stereoselective preference for the R-diastereomers of **35d** and **35f**, catalyzing their hydroxylation with higher efficiency (1.50 μM and 0.81 μM, respectively) compared to the corresponding S-diastereomers. In contrast, the enzyme showed no catalytic activity towards either diastereomer of **35a**, with activity below 0.5 μM [107].

Jones et al. explored the activation and inactivation of vitamin D analogues and prodrugs through metabolic studies. The metabolism of compound **35g** (Figure 9) was examined using HepG2 cells, and the results suggested a lower conversion rate into its metabolites, including **2a** and **35h** (Figure 9), compared to the standard reference compound **35i** (Figure 9). Additionally, the metabolism of **12c** (Figure 9) was evaluated using V79-hCYP24 cells in keratinocyte and hepatoma cell models. Although the data did not provide definitive evidence, the study suggested the formation of saturated side chain metabolites, including C-24 ketone and 23-hydroxylated derivatives of **12c**. The researchers further demonstrated the possible involvement of CYP24 in “vitamin D resistance” associated with **12c** by analyzing the role of CYP24 and other non-vitamin D-related CYP enzymes in the catabolism of this anti-psoriatic agent using cell lines with and without CYP24 expression. The hepatic metabolism of vitamin D analogues was also examined in relation to other factors that may influence their anticancer and antiproliferative properties [108].

Ogawa et al. developed a highly sensitive method for the quantification of trace vitamin D3 metabolites, such as **12b** and **35c** (Figure 9), in urine using liquid chromatography–electrospray ionization tandem mass spectrometry (LC-ESI-MS/MS) with derivatization. Urine samples were treated with β-glucuronidase, followed by purification and derivatization using compound **36** (Figure 9) and its isotope-coded analogue **37** (Figure 9) to enhance detection sensitivity and precision (Figure 6). The developed method achieved a detection limit as low as 0.25 fmol, allowing for the highly sensitive identification of vitamin D3 metabolites. This approach was successfully applied to urine samples, revealing increased metabolite levels following vitamin D3 administration. The method provides a valuable tool for assessing vitamin D status and diagnosing related diseases, with potential applications in both clinical and research settings [109].

### 2.7. Effect of Vitamin D Analogues on Muscular Strength

Xiong et al. (2024) investigated the biological activity of vitamin D derivatives (Figure 10) in enhancing muscle strength and reducing the risk of falls in elderly individuals. A total of 771 participants were recruited for randomized controlled trials (RCTs) assessing the impact of vitamin D analogues on fall incidence, with three trials evaluating compounds **2a** (*n* = 1), **32b** (*n* = 1), and **32a** (*n* = 1). Additionally, 2431 individuals participated in RCTs examining the effects of calcitriol (*n* = 4), **32b** (*n* = 3), and **32a** (*n* = 3) on muscular strength. The findings indicated a 19% reduction in fall incidence when data from the three active vitamin D analogues were combined and analyzed. The pooled and independent analyses revealed no significant impact of any active vitamin D analogue on global muscle strength, hand grip strength, or back extensor strength. In contrast, compounds **32a** and **32b** (Figure 10) demonstrated a significant improvement in quadricep strength in both pooled and independent analyses. Due to insufficient data, the effect of compound **2a** (Figure 10) on quadricep strength could not be evaluated in a separate subgroup analysis. The influence of calcium supplementation on the effects of vitamin D analogues on muscular strength was also examined. Pooled and subgroup analyses of active vitamin D analogues, with or without calcium supplementation, indicated that calcium intake did not significantly alter the observed effects on muscle strength [110].

### 2.8. Activity of Vitamin D Analogues Against Liver Cirrhosis

Reiter et al. investigated the activity of vitamin D analogues and calcitriol in liver cirrhosis through in vitro and in vivo studies. The in vitro antifibrotic properties of compound **2a** (Figure 11) and its analogues were evaluated, revealing a reduction in α-smooth muscle actin (α-SMA) protein expression in mHSCs treated with active vitamin D (**2a**) and its analogues. Compound **32b** (Figure 11) was found to decrease platelet-derived growth factor receptor-β (PDGFR-β) protein expression and contractility in human LX-2 cells stimulated by transforming growth factor-β (TGF-β). Additionally, paricalcitol (**39**, Figure 11), at an equipotent dose to calcitriol (CAL), suppressed TGF-β-induced α-SMA protein expression, as well as ACTA2 and TGF-β mRNA expression. However, no significant effects of vitamin D or its analogues were observed in Kupffer cells. In the in vivo study, carbon tetrachloride (CCl_4_) was used to induce liver fibrosis until quantifiable fibrosis was achieved. Subsequently, compounds **2a** and **39** (Figure 11) were administered to the animals. Mice treated with paricalcitol (PCT) exhibited significantly lower calcium levels compared to untreated mice. In the CCl_4_-induced fibrosis model, both CAL and PCT demonstrated a favorable safety profile, reducing hepatic infiltration of CD11b-positive cells and lowering alanine transaminase (ALT) levels. However, only compound **39**, and not **2a** significantly inhibited fibrosis progression [111].

### 2.9. Vitamin D Analogues Altering the Secretion of Parathyroid Hormone (PTH)

Brown et al. investigated the effects of vitamin D prodrugs, including compounds **40**, **35i**, and **12g** (Figure 12), on the secretion of the parathyroid hormone (PTH) in bovine parathyroid cells. In vitro analysis demonstrated that all three prodrugs reduced PTH secretion by approximately 10% more than compound **2a** (Figure 12), despite having lower predicted activity based on their Vitamin D Receptor (VDR) affinities (0.25% of **2a**). Further metabolic studies of the 1-hydroxy derivatives aimed to identify more active metabolites. The findings suggested the presence of a constitutively active vitamin D-25-hydroxylase enzymes in parathyroid cells, a phenomenon that had not been previously documented. Additionally, the cytochrome P450 inhibitor ketoconazole (50 mM) significantly reduced the metabolism of compound **40** (Figure 12) to undetectable levels. However, ketoconazole had no observable effect on the compound **40**-induced suppression of PTH secretion, indicating that its mechanism of action may be independent of metabolic activation [112].

### 2.10. Vitamin D Analogues with Anti-Psoriasis Activity

Gu et al. reported a synthesis scheme of stereoselective vitamin D derivatives in 2022 and evaluated biological activity such as treatment of psoriasis. (+)-Calcipotriol (**12c**, Figure 13) and its multiple stereo isoform aryls substituted at C20 position were synthesized with the help of retrosynthesis. The synthesized stereoisomer analogues were further evaluated for their cellular efficacy and potency in the treatment of psoriasis by inhibiting the secretion of cytokine IL-17A. Methoxyphenyl-substituted vitamin D derivative (*R*)-**41a** (Figure 13) showed 37.8-fold more potency with IC_50_ value of 12.4 nM as compared to its *S*-form but less potent than **12c** (IC_50_ = 0.29). Similarly, *R* isoform of Compound **41b** (Figure 13) was found to be effective against IL-17A with IC_50_ value of 104 nM while *S*-isoform completely lost its activity. Likewise, R-isoform **41c** (Figure 13) exhibited noticeable inhibition of IL-17A cytokine with an IC_50_ value of 5.90 nM which was 470 times more potent than its S-isoform [113].

### 2.11. Activity of Vitamin D Analogues Against Bronchial Asthma

Thagfan et al. investigated the effects of vitamin D derivatives on patients with bronchial asthma. The selected vitamin D analogues, including compounds **42**, **12i**, **12b**, and **2a** (Figure 14), were evaluated in two groups: the first group consisted of 57 male and 16 female asthmatic patients, while the second group included 31 male and 9 female healthy adults. Serum levels of these compounds were measured and analyzed. Statistical analysis revealed that serum vitamin D levels were significantly lower in asthmatic patients compared to the healthy control group. Additionally, the vitamin D derivatives had a significant impact on key markers of asthma, including forced expiratory volume in one second (FEV₁), forced vital capacity (FVC), FEV₁/FVC ratio, peak expiratory flow (PEF), forced expiratory flow at 25–75% (FEF_25–75%_), eosinophil count, and total immunoglobulin E (IgE) levels. Furthermore, the correlations observed for compounds **12b** and **2a** (Figure 14) were notably stronger than those for compounds **42** and **12i** (Figure 14), suggesting a potentially greater role of **12b** and **2a** in modulating asthma-related parameters [114].

## 3. Summary and Future Perspectives

As discussed in the review study, several researchers have designed, synthesized, and evaluated derivatives of vitamin D. However, a majority of these studies have been confined to evaluating the binding affinity of these derivatives with the Vitamin D Receptor (VDR), their transcriptional activity, and their role in calcium homeostasis. Although derivatives have also been explored for activities such as anti-proliferative, anti-osteoporosis, anti-fibrotic, anti-inflammatory, anti-asthmatic effects, immune regulation, and liver fibrosis, their metabolic fate remains comparatively less studied.

One crucial aspect that has received limited attention in these studies is the metabolic pathway of vitamin D derivatives. The metabolism of vitamin D is well established, involving activation and subsequent deactivation steps. Hydroxylation in the liver by CYP2R1 (microsomal) and CYP27A1 (mitochondrial) enzymes converts vitamin D_3_ into 25-hydroxyvitamin D_3_ (25-OH-D_3_), which is further hydroxylated in the kidney by CYP27B1, producing the biologically active 1,25-dihydroxyvitamin D_3_ (1α,25-(OH)_2_D_3_) [115,116].

Both 25-OH-D_3_ and 1,25-(OH)_2_D_3_ undergo inactivation primarily by CYP24A1, which hydroxylates these forms. CYP24A1 converts 1,25-(OH)_2_D_3_ into 24R,25-(OH)_2_D_3_ and further degrades it through two distinct pathways: the C23 lactone pathway, leading to the formation of 1,25-(OH)_2_D_3_-26,23-lactone, and the C24 oxidation pathway, which generates calcitroic acid, subsequently excreted via bile. These enzymatic transformations tightly regulate vitamin D levels, ensuring calcium homeostasis and physiological balance while preventing toxicity [117].

The levels of vitamin D-metabolizing enzymes, such as CYP24A1, along the metabolic cascade can disrupt the proper balance of metabolites and have been reported to contribute to vitamin D-related complications. Studies have shown that elevated CYP24A1 levels are associated with vitamin D insufficiency and resistance to vitamin D therapy, particularly in chronic kidney disease (CKD). Additionally, the inhibition of CYP24A1 has been documented as a promising strategy for improving vitamin D levels and treating secondary hyperparathyroidism in CKD [117].

The majority of newly investigated vitamin D analogues lack comprehensive metabolic inactivation studies, which are crucial for discovering promising lead molecules. Targeting enzymes such as CYP24A1 could serve as a therapeutic approach by developing inhibitors or activators, ultimately influencing the levels of vitamin D and its metabolites.

## 4. Conclusions

Vitamin D and its analogues have demonstrated significant therapeutic potential across various disease conditions, including cancer, autoimmune disorders, inflammatory diseases, liver fibrosis, and metabolic disorders. Structural modifications in the vitamin D3 skeleton have led to the development of synthetic derivatives with optimized receptor selectivity, improved pharmacokinetic and pharmacodynamic properties, and reduced systemic toxicity. These derivatives act as modulators of the Vitamin D Receptor (VDR), displaying both agonistic and antagonistic effects, thereby influencing key cellular pathways involved in disease pathophysiology.

Apart from a number of studies focused on cancer, several studies have reported superior efficacy analogues compared to calcitriol in modulating transcriptional activity, suppressing nuclear NF-κB-p65, and regulating cytokines such as IL-17 and IκB-α, indicating their strong anti-inflammatory potential. Additionally, vitamin D derivatives have been found to be effective against IBD, osteoclast inhibition, improving muscle strength and lung function. Moreover, vitamin D analogues and calcitriol have exhibited hepatoprotective effects and anti-psoriasis activity by downregulating IL-17A secretion. The review highlights the potential of vitamin D derivatives as effective therapeutic agents and emphasizes the need for further research to develop clinically approved compounds with minimal impact on calcium–phosphorus homeostasis while maximizing therapeutic efficacy.

## Figures and Tables

**Figure 1 biomedicines-13-01002-f001:**
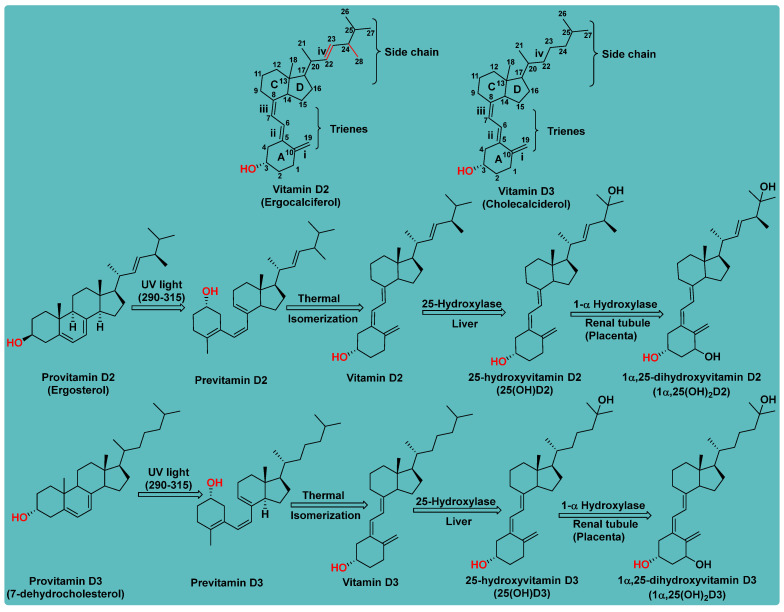
Conversion of inactive forms of Vitamin D to their respective active forms.

**Figure 2 biomedicines-13-01002-f002:**
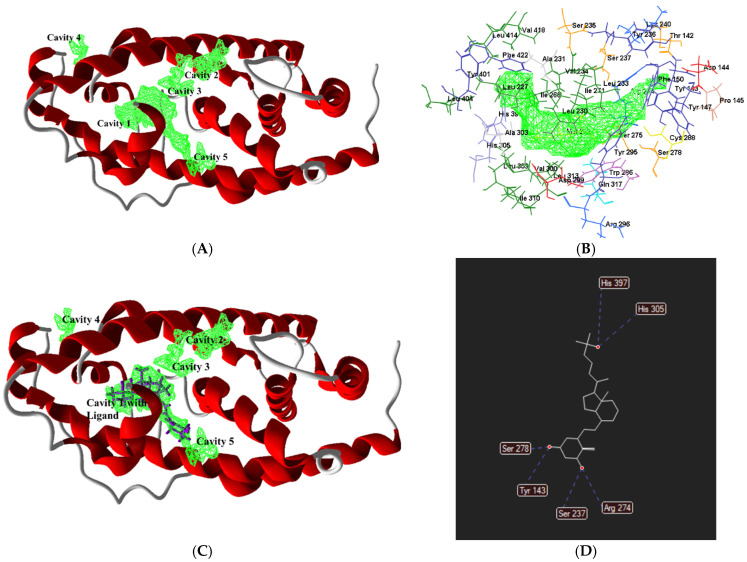
Molecular simulation of vitamin D receptor, (**A**) secondary structure of protein with cavity, (**B**) amino acid around to the cavity 1, (**C**) protein with bounded ligand, (**D**) interaction of calcitriol ligand with active site of VDR).

**Figure 3 biomedicines-13-01002-f003:**
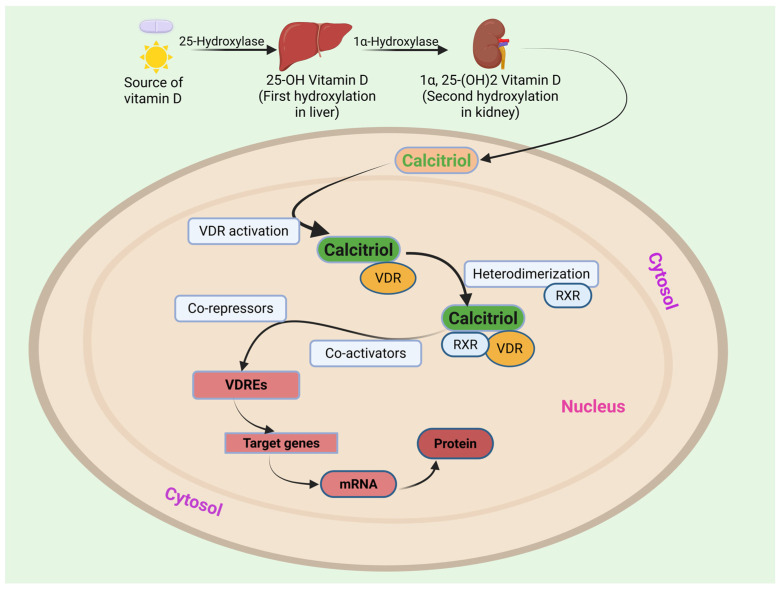
Sequential activation of the Vitamin D and its mechanism of action.

**Figure 4 biomedicines-13-01002-f004:**
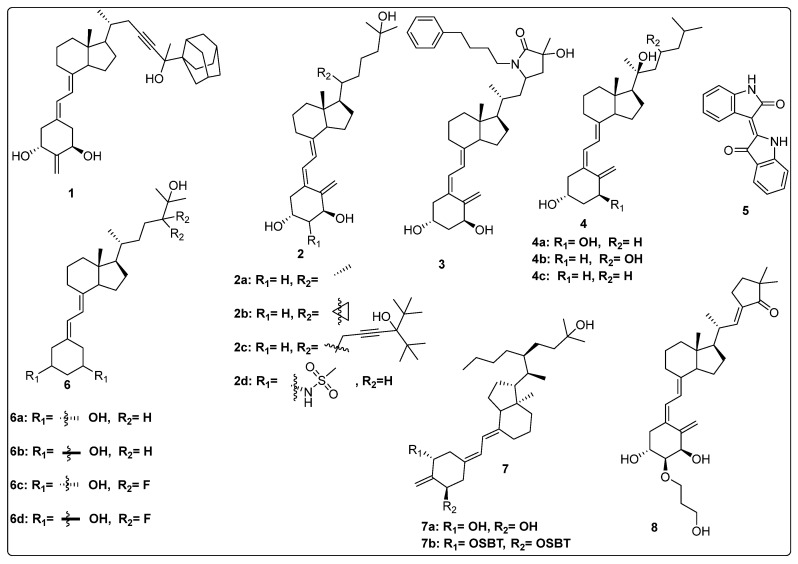
Structures of vitamin D derivatives and their binding affinity towards VDR.

**Figure 5 biomedicines-13-01002-f005:**
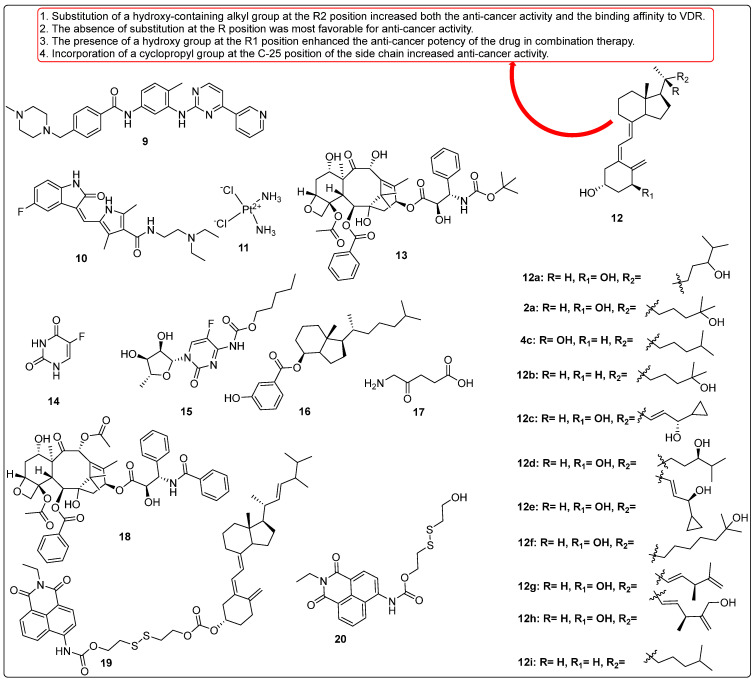
Vitamin D derivatives as anti-cancer agents with their SAR study.

**Figure 6 biomedicines-13-01002-f006:**
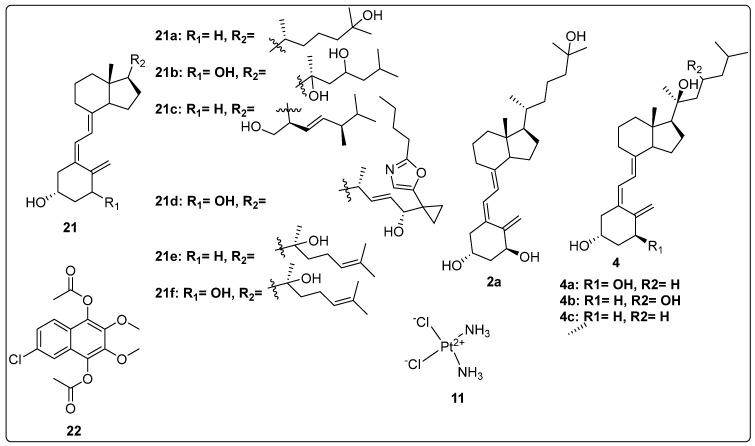
Vitamin D derivatives as anti-inflammatory agents.

**Figure 7 biomedicines-13-01002-f007:**
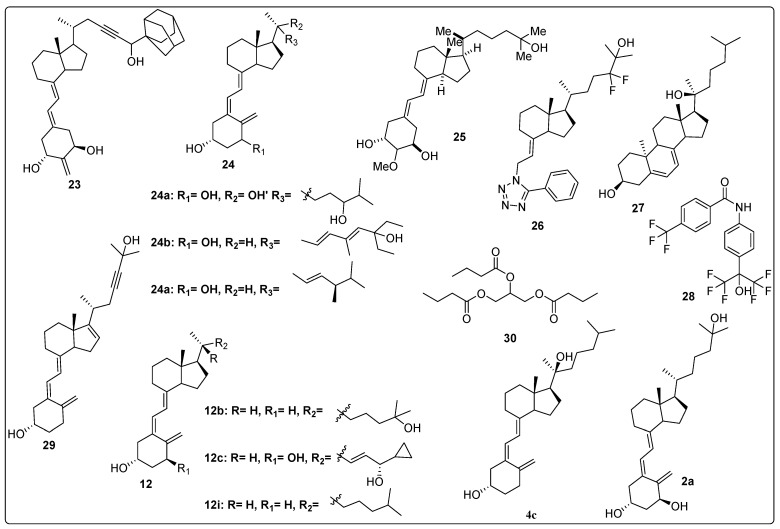
Vitamin D analogues explored for VDR binding affinities.

**Figure 8 biomedicines-13-01002-f008:**
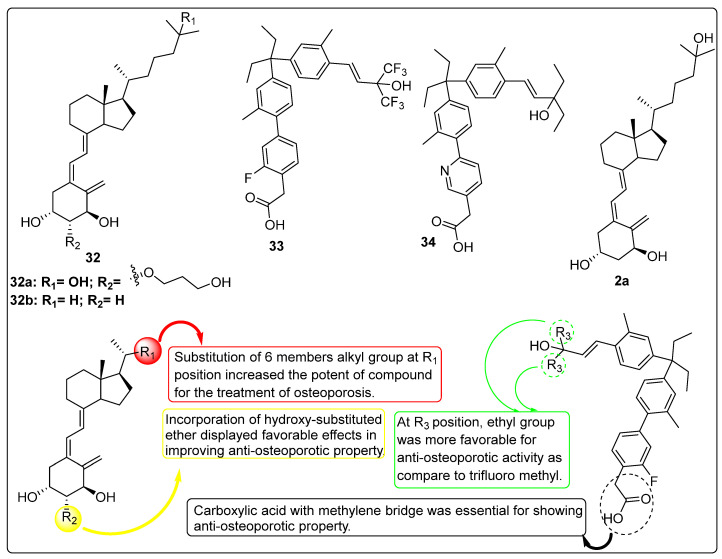
Vitamin D derivatives with anti-osteoporotic properties and SAR study.

**Figure 9 biomedicines-13-01002-f009:**
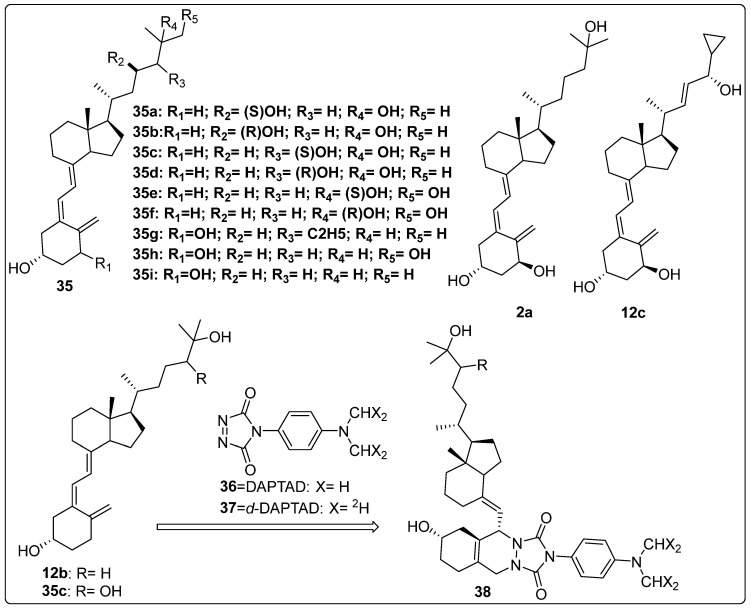
Vitamin D derivatives studied for metabolic stability.

**Figure 10 biomedicines-13-01002-f010:**
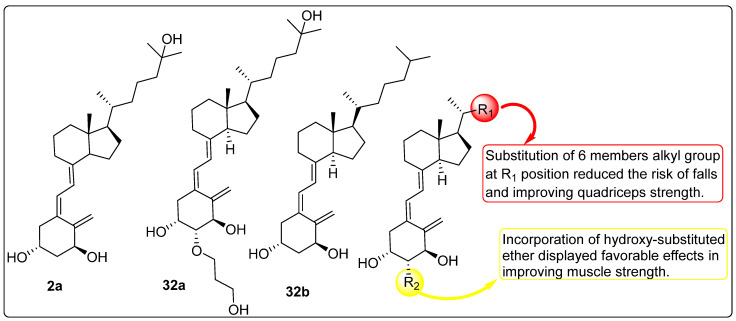
Vitamin D analogues with their muscle strength property and SAR study.

**Figure 11 biomedicines-13-01002-f011:**
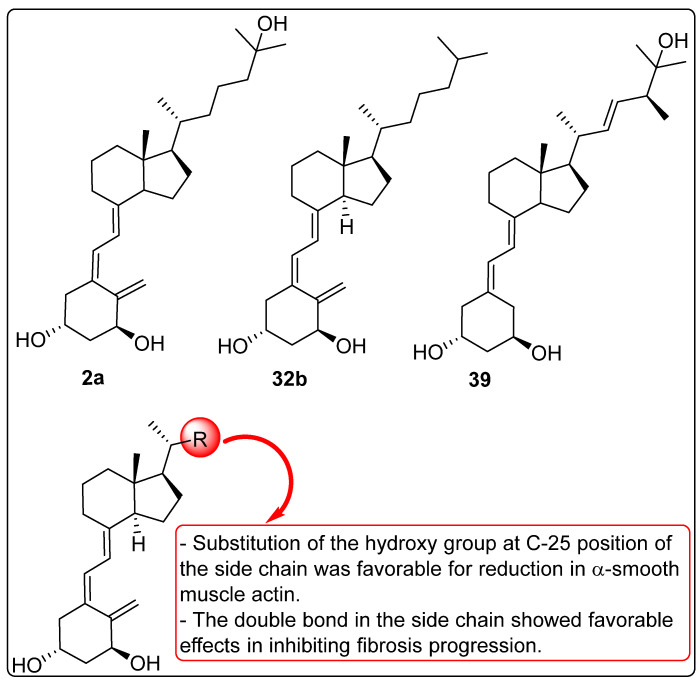
Vitamin D analogues used in treatment of liver cirrhosis, along with SAR study.

**Figure 12 biomedicines-13-01002-f012:**
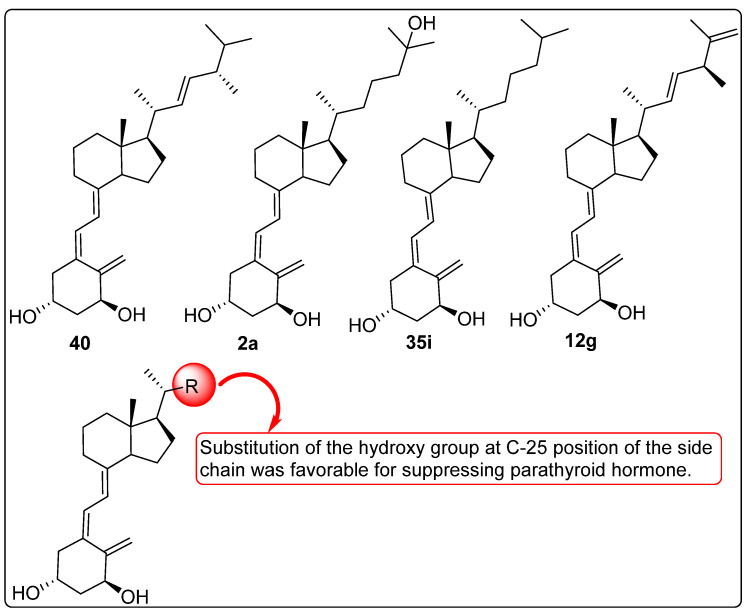
Anti-hyperthyroidism and SAR study of vitamin D analogues.

**Figure 13 biomedicines-13-01002-f013:**
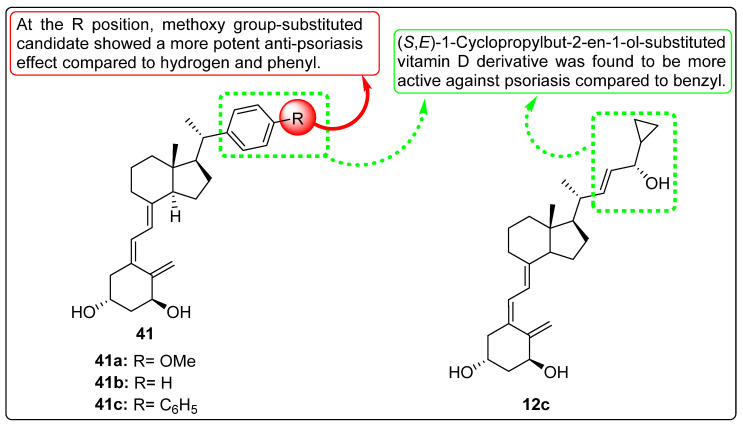
Vitamin D analogues with anti-psoriasis activity and SAR.

**Figure 14 biomedicines-13-01002-f014:**
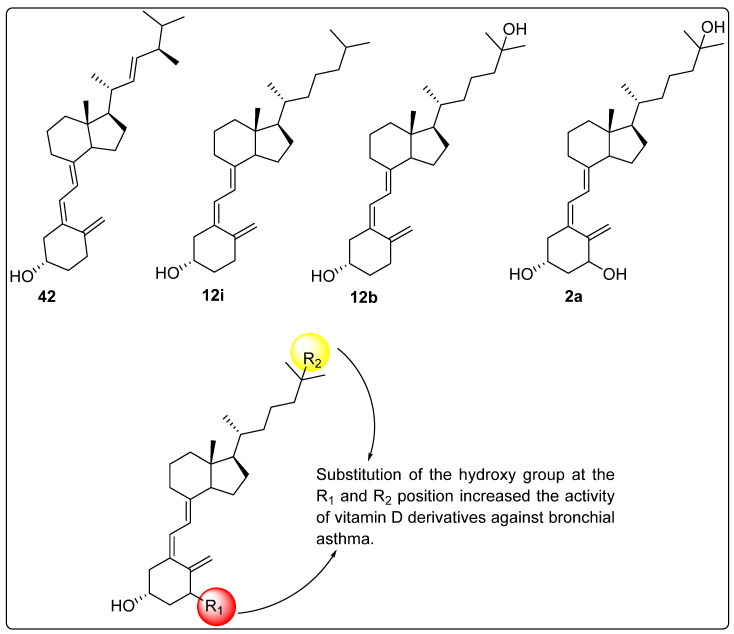
Vitamin D analogues with anti-asthmatic activity and SAR.

**Table 1 biomedicines-13-01002-t001:** Characteristics of vitamin D receptor cavities.

S. No.	Cavity	Position	Volume (A^3^)	Surface (A^2^)
1	Cavity 1	10.692, 23.1393, 33.9638	202.752	437.76
2	Cavity 2	3.6615, 26.5, 49.2174	25.6	110.08
3	Cavity 3	1.33962, 25.9334, 45.0743	17.408	83.2
4	Cavity 4	3.9975, 36.356, 303373	12.8	61.44
5	Cavity 5	0.797499, 14.756, 43.8733	12.8	52.48

**Table 2 biomedicines-13-01002-t002:** Different natural and synthetic forms of vitamin D clinically used against diverse indications.

Form of Vitamin D	Name	Structure	Uses
Vitamin D2	Ergocalciferol	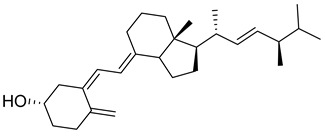	Treatment of vitamin D deficiency, management of hypoparathyroidism, used in fortified foods and supplements.
Vitamin D3	Cholecalciferol	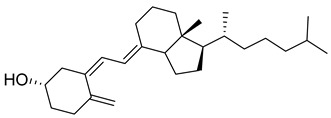	Treatment/prevention of vitamin D deficiency, rickets, and osteomalacia, used in fortified foods and supplements.
25-Hydroxyvitamin D3	Calcifediol	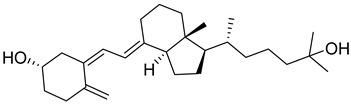	Treatment of severe vitamin D deficiency, especially in patients with liver dysfunction.
1,25 Dihydroxyvitamin D3	Calcitriol	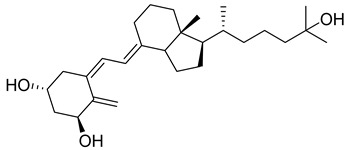	Treatment of hypocalcemia, secondary hyperparathyroidism, osteoporosis, chronic kidney disease (CKD), and renal osteodystrophy.
1-Hydroxyvitamin D3	Alfacalcidol	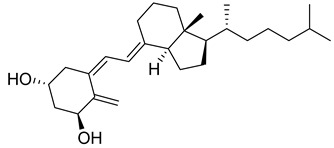	Management of CKD-associated bone disease, reversal of myopathy, hypoparathyroidism, and other calcium-related disorders.
1α-Hydroxyvitamin D2	Doxercalciferol	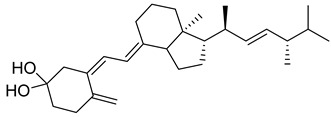	Treatment of secondary hyperparathyroidism in CKD.
Synthetic Analogue	Paricalcitol	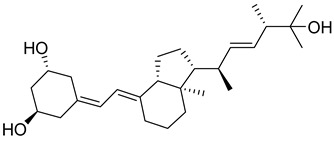	Treatment of secondary hyperparathyroidism in CKD without causing hypercalcemia.
Synthetic Vitamin D3 analogue	Eldecalcitol	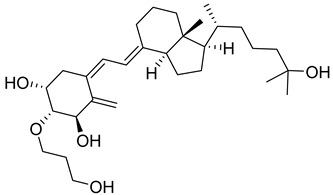	Treatment of osteoporosis; enhances calcium absorption in bones.
Synthetic Analogue	Calcipotriol (Calcipotriene)	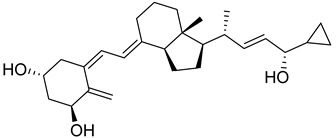	Treatment of psoriasis; regulates keratinocyte differentiation and proliferation.
Synthetic Analogue	Maxacalcitol	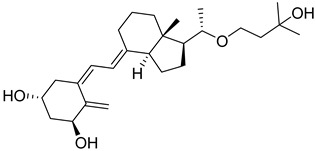	Treatment of secondary hyperparathyroidism; management of psoriasis (in specific regions).
Fluorinated Synthetic Analogue	Falecalcitriol	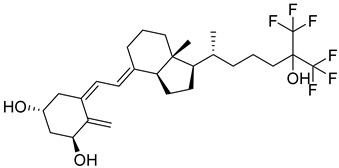	Used for suppressing PTH in patients having chronic renal failure.
Metabolite of 7-dehydrocholesterol	Lumisterol	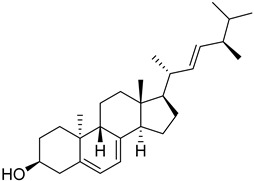	Used for anti-aging and photoprotection activity.
Vitamin D4	22-dihydroergocalciferol	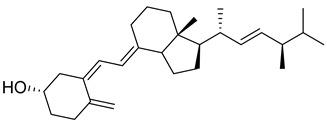	Research studies for receptor binding and prevention of cancer and cardiovascular disease.
Vitamin D5	Sitocalciferol	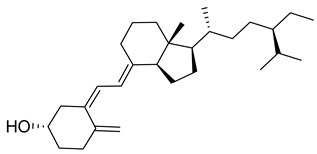	Explored for research in the area of oncology and plant-based nutrition

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
