# Peer review of "Recent Advancements Towards the Use of Vitamin D Isoforms and the Development of Their Synthetic Analogues as New Therapeutics"

_biomedicines, 2025, doi:10.3390/biomedicines13041002_

Round 1

Reviewer 1 Report

Comments and Suggestions for Authors

The authors provide an informative and updated overview on the pharmacology of natural vitamin D isoforms and their synthetic analogues aiming the development of new therapeutics. The title “Comprehensive Review on Chemistry and Pharmacology of Vitamin D Isoform and Derivatives from Natural Origin to Synthetic Derivatives” is somewhat misleading, since considered as a “comprehensive” review it lacks the groundbreaking work accomplished in the past decades from Uskokovic, De Luca, Posner and many others in the field. Additionally, almost no “chemistry” is described in the article, neither exactly how the compounds are designed nor how they are synthesized. However, describing the synthesis of VD analogues might not really be essential, since it could be looked up from the interested reader in the references. Therefore I may suggest to modify/reconsider the title in the sense of “Recent advancements towards the use of vitamin D isoforms and the development of their synthetic analogs as new therapeutics”. 

I also recommend to reconsider the keywords. They are either too general (“anti-cancer”) and/or too long (“Mechanism…”).

Despite enormous efforts in the past decades, the discovery of drugs based on vitamin D has been very limited and actually ended up disappointingly.  The true reasons for this failure would have to be mentioned in the introduction and/or the abstract, i.e. toxicity due to calcemic effects (such as in case of calcitriol), leading to hypercalcemia, at least at high dosage, and rapid metabolization. These two drawbacks were tried to overcome by the design and synthesis of thousands of analogs, although without substantial success. Much effort was concentrating on the improvement of potency just by optimizing VDR affinity (experimentally and by computational docking studies, reviewed by Maestro), what is also extensively described in the manuscript, but this approach has been shown not to overcome the mentioned drawbacks. Enzyme defects (i.e. CYP24A1) along the cascade of metabolization may also disturb the proper balance/ratio of metabolites and cause various diseases (i.e. 1,25(OH)2D3/24,25(OH)2D3-ratio), what is important to be considered for a differentiated diagnosis of VD dependent diseases. These enzyme defects may serve as a starting point for a therapeutic approach to develop enzyme inhibitors or activators (i.e. Posner: CYP24A1 inhibitors). Results/Outcome of pharmacological studies are sometimes not explained/discussed accordingly (sections 2.2 to 2.11). Some structure-activity relationship (apart from VDR-binding) and pharmacokinetic considerations would be interesting to know in more detail. How far are structures rationally designed or do results rely on try-and error? This is particularly true for (not obvious) combination of drugs with VD analogs.

I recommend to draw all structures consistently with CD-ring in plane (Fig´s 1, 4, 5, 6, 7, 8, 9, 10, 11, 12, 13 and 14.

Introduction: Fig.1: wrong configuration of C-3 in structure of Previtamin D2 (3-OH); Vitamin D2, 25-Hyroxyvitamin D2, 1,25-dihydroxyvitamin D2: all have wrong configuration of double bond at C-6; 1a should be drawn/shown correctly in structures of 1a,25(OH)2D2 and 1a,25(OH)2D2; the authors should pay attention of consistent use of capital letters

Table 2.: I recommend to display the corresponding structures in a separate column of the table

Section 2.1 overlap with 2.4 and should be restructured

The text is a bit hard to read due to many spelling and grammar errors.

Some minor corrections along the text:

Line

replace

by

comments

18

analogues

metabolites

24

for

against

70

1,25-dihydroxyvitamin D (1,25(OH)2D)

1,25-dihydroxyvitamin D (1,25(OH)2D), calcitriol

72

require

requires

73

the

its

81

Mention calcitriol

87

It is active metabolites

The active metabolite of ergosterol is ..

107

To be preferred

To be the preferred

132

Vitamin D Receptor..

The Vitamin D Receptor…

138

kidney

kidneys

138

Structure of Vitamin D receptor comprising of

The structure of the Vitamin D Receptor comprises of..

157

The calcitriol was found to be completely fit in active

site of vitamin D receptor

Calcitriol was found to completely fit in the active site of the vitamin D receptor

171-175

Was already mentioned in the introduction and should be shifted or deleted

189

Oral Cholecalciferol is effective in addressing the vitamin D deficiency.

Orally administered cholecalciferol is effective in addressing vitamin D deficiency.

198

require

required

198

Calcium, Phosphorus

calcium, phosphorus

207

Alfacalcidol is analog of vitamin D, act

Alfacalcidol is an analog of vitamin D3, that act

214

Paricalcitol an analog of 1,25-dihydroxyergocalciferol used

Paricalcitol, an analog of 1,25-dihydroxyergocalciferol, is used

218

Drug is effectively in treating

The drug is effectively treating

222

drugs of psoriasis

drugs for treatment of psoriasis

226

Animal studies evident it a potent inhibitor

Animal studies have shown that it is a potent inhibitor

228

six fluorine substitution in place of six hydrogens

six fluorine atoms replacing six hydrogens

230

Falecalcitrio

Falecalcitriol

241

by the UV radiation.

by UV radiation.

Vitamin D4 is reported in limited studies in some mushrooms and endophytic fungi often present in

plants

Vitamin D4 is reported in limited studies to be found in some mushrooms and endophytic fungi, in turn often present in

plants

348

cyp1A1 and cyp1B1

Size too big

425

14

14

427

individual treatments

using only …

432

Fig.3

Fig.5

444

vitamin analogue.

A vitamin analogue.

445

vitamin analogue

vitamin analogues

487

the anticancer of metabolite

the anticancer activity of metabolite

502

16

16

547

with combination

in combination

575

[66]

[66].

643

The finding of study demonstrated

The findings of this study demonstrated

652, 654

compound 26

compound 26

655

26

26

667, 668, 669

27, 28, 4c, 2a, 27, 28

Size too big

725

2a

2a

754

2a

2a

818

34

34

1016, 1191

d

D

References: For most references doi and/or PMID is missing

Summing up, the manuscript is presumably of significant interest for numerous readers/scientists  working in the field and who are aiming to update their knowledge without examining the enormous wealth of studies recently published, and merits to be published in biomedicines after minor corrections.

Comments on the Quality of English Language

The text is a bit hard to read due to many spelling and grammar errors.

Author Response

 The detailed justifications in response to queries/comments are provided below:

Reviewer 1:

  1. The authors provide an informative and updated overview on the pharmacology of natural vitamin D isoforms and their synthetic analogues aiming the development of new therapeutics. The title “Comprehensive Review on Chemistry and Pharmacology of Vitamin D Isoform and Derivatives from Natural Origin to Synthetic Derivatives” is somewhat misleading, since considered as a “comprehensive” review it lacks the groundbreaking work accomplished in the past decades from Uskokovic, De Luca, Posner and many others in the field. Additionally, almost no “chemistry” is described in the article, neither exactly how the compounds are designed nor how they are synthesized. However, describing the synthesis of VD analogues might not really be essential, since it could be looked up from the interested reader in the references. Therefore, I may suggest to modify/reconsider the title in the sense of “Recent advancements towards the use of vitamin D isoforms and the development of their synthetic analogs as new therapeutics”.

Response: We sincerely appreciate your suggestion. After careful consideration, we have modified the title of the paper accordingly.

  1. I also recommend to reconsider the keywords. They are either too general (“anti-cancer”) and/or too long (“Mechanism…”).

Response: The keywords have been revised as per the suggestion.

  1. Despite enormous efforts in the past decades, the discovery of drugs based on vitamin D has been very limited and actually ended up disappointingly. The true reasons for this failure would have to be mentioned in the introduction and/or the abstract, i.e. toxicity due to calcemic effects (such as in case of calcitriol), leading to hypercalcemia, at least at high dosage, and rapid metabolization. These two drawbacks were tried to overcome by the design and synthesis of thousands of analogs, although without substantial success. Much effort was concentrating on the improvement of potency just by optimizing VDR affinity (experimentally and by computational docking studies, reviewed by Maestro), what is also extensively described in the manuscript, but this approach has been shown not to overcome the mentioned drawbacks. Enzyme defects (i.e. CYP24A1) along the cascade of metabolization may also disturb the proper balance/ratio of metabolites and cause various diseases (i.e. 1,25(OH)2D3/24,25(OH)2D3-ratio), what is important to be considered for a differentiated diagnosis of VD dependent diseases. These enzyme defects may serve as a starting point for a therapeutic approach to develop enzyme inhibitors or activators (i.e. Posner: CYP24A1 inhibitors). Results/Outcome of pharmacological studies are sometimes not explained/discussed accordingly (sections 2.2 to 2.11). Some structure-activity relationship (apart from VDR-binding) and pharmacokinetic considerations would be interesting to know in more detail. How far are structures rationally designed or do results rely on try-and error? This is particularly true for (not obvious) combination of drugs with VD analogs.

Response:  Reviewer rightly stated that the prime reasons for the failure of new vitamin D based drugs are their toxicity due to calcemic effects and undesirable metabolic profile. These parts we already mentioned in the discussion part and now as per the suggestion we also included in the abstract of the manuscript.

     Reviewer mentioned very crucial aspects related to metabolism, specially inactivation of vitamin D analogues which ultimately decide the balance/ratio of metabolites and can cause diseases related to vitamin D. This aspect is not studied for the majority of the newly investigated molecules reported by various researchers. Considering the importance of point, dedicated paragraph has been included at end the manuscript, focusing on metabolism of vitamin D, specially on roles of enzymes like CYP2R1, CYP27A1 and CYP24A1.

     Results/Outcome of pharmacological studies of sections 2.2 to 2.11 have been reviewed and modification are made at the suitable positions.

    Structure-activity relationship (SAR) pertaining to the binding affinity towards the VDR are included in the manuscript. To include SAR studies for the pharmacokinetic perspective, data of analogues library required focussing on the absorption, distribution, metabolism and excretion, however such comprehensive studies are not covered in majority of the research papers, so pharmacokinetic based SAR is not included, rather finding of individual studies are discussed, wherever such data are mentioned by the researchers. 

     Talking of about the rationality of the design, in most of reported studies researchers play around the structure modification of vitamin D core scaffold, which can be covered under the ligand-based drug designing and optimization. However, in few studies, computational studies are also performed but such studies are just performed in order to investigate the putative binding mode of molecules at the receptor site. 

  1. I recommend to draw all structures consistently with CD-ring in plane (Fig´s 1, 4, 5, 6, 7, 8, 9, 10, 11, 12, 13 and 14).

Response: We are grateful to you for guiding about structural errors, we have re-drawn the structures as per the suggestion.

  1. Introduction: Fig.1: wrong configuration of C-3 in structure of Previtamin D2 (3-OH); Vitamin D2, 25-Hyroxyvitamin D2, 1,25-dihydroxyvitamin D2: all have wrong configuration of double bond at C-6; 1a should be drawn/shown correctly in structures of 1a,25(OH)2D2 and 1a,25(OH)2D2; the authors should pay attention of consistent use of capital letters.

Response: We are thankful for notifying the errors. We have corrected the mistake in the representative structure as per your suggestion.

  1. Table 2.: I recommend to display the corresponding structures in a separate column of the table.

Response: As per suggestion, we incorporated the corresponding structures of all compounds in a separate column of Table 2.

  1. Section 2.1 overlap with 2.4 and should be restructured.

Response: We sincerely acknowledge our mistake; it was a typographical error in Section 2.4 during drafting. The headings are distinct from each other—Section 2.1 is correct and represents only the binding affinity of vitamin D derivatives toward the vitamin D receptor. However, Section 2.4 was incorrect, as it should display the dual activities of vitamin D derivatives, including both binding affinity and anticancer properties. We have now corrected the heading of Section 2.4.

  1. Some minor corrections along the text:

Response: As per your advice, we have corrected the text at the corresponding position.

  1. References: For most references doi and/or PMID is missing.

Response: As per your advice, we have incorporated DOI and/or PMID in the corresponding references and added some additional references.

Reviewer 2 Report

Comments and Suggestions for Authors

Review is well written and there is good coverage of the synthetic analogs of Vitamin D. To avoid any confusion authors should modify the analogs to correct hydroxy derivatives. As mostly 1,25 analogs are relevant to VDR. For example, in section 2.7 and 2.8 we should call in the headings in the text dihydroxy Vitamin D versus just just vitamin D.

Author Response

 The detailed justifications in response to queries/comments are provided below:

Reviewer 2:

  1. Review is well written and there is good coverage of the synthetic analogs of Vitamin D. To avoid any confusion authors should modify the analogs to correct hydroxy derivatives. As mostly 1,25 analogs are relevant to VDR. For example, in section 2.7 and 2.8 we should call in the headings in the text dihydroxy Vitamin D versus just just vitamin D.

Response: Thank you for your suggestion. However, we cannot replace "Vitamin D" with "dihydroxy Vitamin D" in Sections 2.7 and 2.8, as the position and number of hydroxy groups vary within these sections. All forms of Vitamin D contain a hydroxy group at the C-3 position. For instance, in Fig. 10 (Section 2.7), compound 32a has dihydroxy groups at the 1α and C-25 positions of Vitamin D, along with a 3-(oxidaneyl)propan-1-ol at C-2, whereas compound 32b has only a single hydroxy group at the 1α position. The same reasoning applies to Section 2.8.
